# Glacial isostatic uplift of the European Alps

Jürgen Mey[1], Dirk Scherler[2,3], Andrew D. Wickert[4], David L. Egholm[5], Magdala Tesauro[6], Taylor F. Schildgen[1,2] & Manfred R. Strecker[1]

Following the last glacial maximum (LGM), the demise of continental ice sheets induced crustal rebound in tectonically stable regions of North America and Scandinavia that is still ongoing. Unlike the ice sheets, the Alpine ice cap developed in an orogen where the measured uplift is potentially attributed to tectonic shortening, lithospheric delamination and unloading due to deglaciation and erosion. Here we show that ~90% of the geodetically measured rock uplift in the Alps can be explained by the Earth's viscoelastic response to LGM deglaciation. We modelled rock uplift by reconstructing the Alpine ice cap, while accounting for postglacial erosion, sediment deposition and spatial variations in lithospheric rigidity. Clusters of excessive uplift in the Rhône Valley and in the Eastern Alps delineate regions potentially affected by mantle processes, crustal heterogeneity and active tectonics. Our study shows that even small LGM ice caps can dominate present-day rock uplift in tectonically active regions.

[1] Institut für Erd- und Umweltwissenschaften, Universität Potsdam, 14476 Potsdam, Germany. [2] Helmholtz Centre Potsdam, GFZ German Research Centre for Geosciences, Telegrafenberg, 14473 Potsdam, Germany. [3] Institute of Geological Sciences, Freie Universität Berlin, 12249 Berlin, Germany. [4] Department of Earth Sciences and Saint Anthony Falls Laboratory, University of Minnesota, Minneapolis, 55455 Minnesota, USA. [5] Department of Geoscience, Aarhus University, 8000 Aarhus, Denmark. [6] Department of Earth Sciences, Utrecht University, 3508 Utrecht, Netherlands. Correspondence and requests for materials should be addressed to J.M. (email: mey@geo.uni-potsdam.de).

Recent vertical movements of the Earth's crust are mostly due to tectonic deformation along plate boundaries, volcanism and changes in crustal loading from water, ice and sediments[1]. The decay of continental ice sheets caused uplift of the formerly glaciated regions and was the primary cause for the Holocene eustatic sea level rise, which is one of the main concerns of the impacts of global warming on coastal communities worldwide[2]. Changes in the ice load of tectonically active mountain ranges, such as the Alps, the Alaska Range or the Himalaya, although much smaller, nevertheless trigger an isostatic response. The induced surface uplift and/or subsidence is thought to have caused changes in fluvial networks[3], and the resulting stress changes in the Earth's crust can influence crustal deformation and seismicity[4] and might have triggered some of the largest intraplate earthquakes since

last glacial maximum (LGM) deglaciation[5]. The key controls on how the Earth responds to changes in crustal loading are the viscosity of the upper mantle and the lithospheric effective elastic thickness (EET)—a geometric measure of the flexural rigidity of the lithosphere, which describes the resistance to bending under the application of vertical loads[1]. Most previous estimates of mantle viscosity come from old and tectonically stable continents, where the vertical motion can almost entirely be attributed to postglacial rebound[6]. In contrast, the complexity of the uplift signal in tectonically active orogens requires the relative contribution of different potential driving mechanisms to be disentangled.

For half a century, the cause for recent uplift of the European Alps has been debated. Possible drivers of uplift include postglacial rebound[7], erosional unloading[8], tectonic deformation[9], lithospheric slab dynamics[10] and combinations thereof (Fig. 1). Some of these processes, such as lithospheric delamination, manifest themselves on timescales of $\sim 10^6$–$10^7$ years, whereas others, such as postglacial rebound, occur relatively rapidly ($\sim 10^3$ years). New approaches to modelling orogen-scale sediment storage[11], glaciation[12] and spatial variations in EET (ref. 13) provide new constraints for estimating the contribution of glacial isostatic adjustment (GIA) to present-day uplift rates in the European Alps.

Mountain building in the European Alps is due to the convergence of Africa and Eurasia beginning in the Mesozoic with continental collision culminating in the Eo-Oligocene[14]. A late phase of outward tectonic growth in the Early Miocene created the Jura Mountains and thrusting of the Swiss Plateau[14] (Fig. 2). Further tectonic shortening was accompanied by eastward extrusion of the Eastern Alps and exhumation of metamorphic domes in the Central Alps[15]. The cessation of outward tectonic expansion of the Western and Central Alps during the Late Miocene might reflect an increase in the ratio of erosional to accretionary material flux and the onset of orogenic decay[16]. During the Pleistocene, the Alps were repeatedly

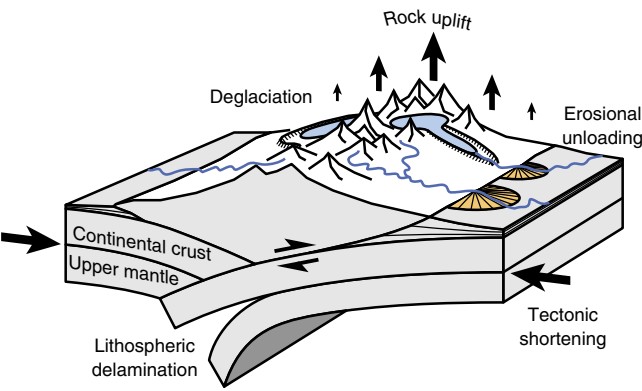

**Figure 1 | Processes contributing to rock uplift in a contractional orogen.** The individual components are interdependent and their relative contribution to rock uplift changes over time. Blue and orange polygons indicate glaciers and alluvial fans, respectively.

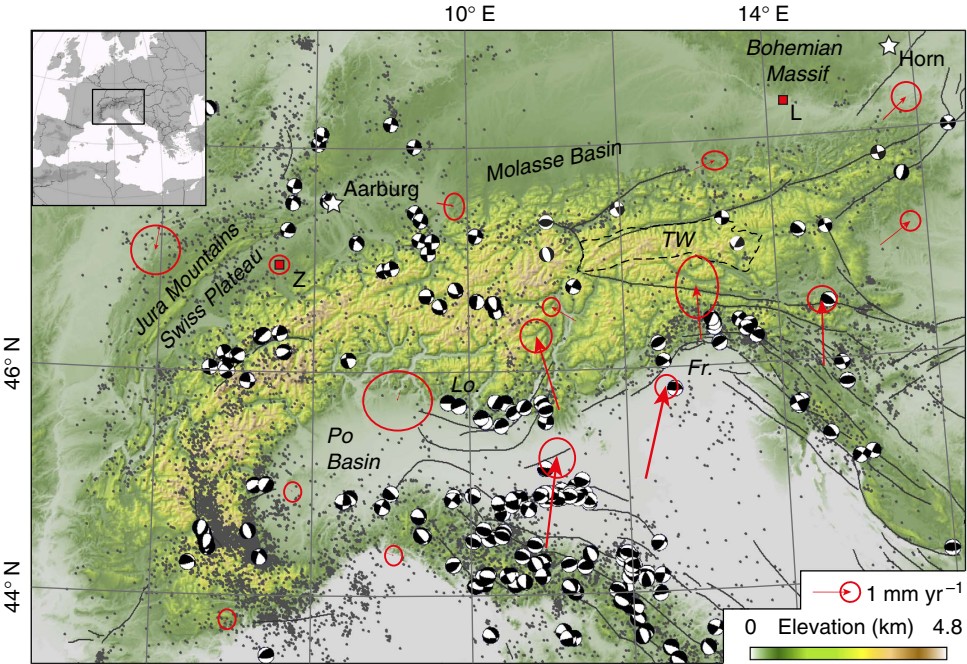

**Figure 2 | Seismotectonic setting.** Seismicity (grey dots, NEIC, 1973–2008), focal plane solutions[63] and seismogenic faults (black solid lines, http://diss.rm.ingv.it/share-edsf/) superimposed over a DEM of the study area. Red arrows depict the horizontal velocity field of permanent GPS stations in a Europe-fixed reference frame[64]. Error ellipses show 1-sigma (67%) confidence level. Stars indicate locations of the reference points for the Swiss and Austrian precise levelling data, respectively. 'L' and 'Z' are the locations of the permanent GPS stations used to adjust the levelling data to the global reference frame IGb08. Fr., Friuli; Lo., Lombardy; TW, Tauern Window.

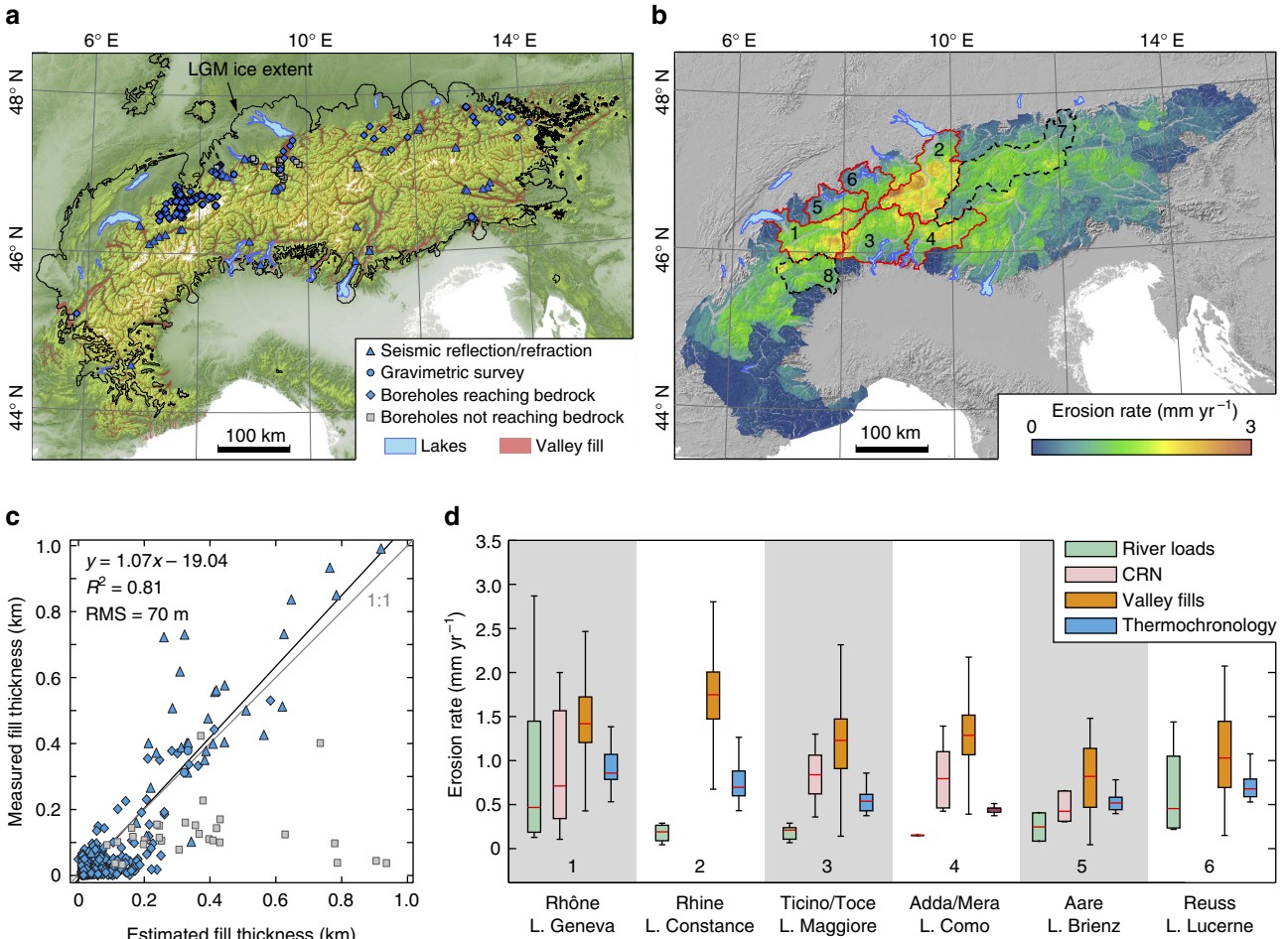

**Figure 3 | Valley-fill thickness and postglacial erosion rates.** (**a**) Present-day ice cover[65] (white), LGM ice extent[31] (black outline), distribution of sedimentary valley fills and locations of fill-thickness measurements (see legend). (**b**) Map of postglacial erosion rates derived from the sediment distribution in **a** and an additional 10% of exported material (see text for details). Data from catchments 1 to 6 (highlighted with red boundaries) are provided in **d**. The eroded mass in catchments 7 and 8 was manually increased to smooth abrupt changes in erosion rates across the corresponding basin boundaries. (**c**) Comparison of estimated and measured valley-fill thicknesses (see key in panel a for symbol details). See Supplementary Table 1 for data sources. (**d**) Comparison of the derived erosion rates with data from modern river loads[26], cosmogenic nuclides[27–29] and thermochronology[18]. On each box, the central red mark is the median, the edges of the box are the 25th and 75th percentiles and the whiskers extend to the minimum and maximum values. Rivers and corresponding marginal lakes are indicated. L., Lake.

glaciated with ice caps that covered almost the entire mountain belt and substantial parts of the northern foreland[17]. Locally, glaciation was presumably associated with a twofold increase in exhumation rates[18,19] and topographic relief[20], which may be controlled by feedbacks between glacial erosion, crustal unloading, isostatic uplift and deep-seated processes.

Permanent global positioning system (GPS) stations indicate ongoing crustal convergence of 1–2 mm yr$^{-1}$ across the Eastern Alps (Fig. 2) that is controlled by the counterclockwise rotation of the Adriatic plate[21]. The convergence is accommodated by thrusting in the Italian Friuli and Lombardy regions and by eastward extrusion along strike-slip faults[15,21]. In the Central and Western Alps, however, only minor or no crustal shortening can be detected[22] and earthquake focal plane solutions are dominated by extensional and strike-slip mechanisms (Fig. 2).

In this study, we re-evaluate the effect of GIA on the present-day rock uplift in the Alps while accounting for postglacial erosion, sediment deposition and variations in lithospheric strength. We show that most of the postglacially eroded material was trapped within the mountain belt and did not contribute to erosional unloading as previously suggested[8]. Instead our results demonstrate that the long-wavelength uplift signal is best

explained by the Earth's viscoelastic response to ice unloading after the LGM. We conclude that present-day uplift rates in other tectonically active and glaciated mountain belts could also carry a component related to LGM deglaciation.

## Results

**Alpine valley fills and postglacial erosion.** Among the most prominent features of the Alpine landscape are overdeepened valleys that were carved by glaciers and are now partially buried by thick sedimentary deposits. The isostatic adjustment to deglaciation was likely attenuated by the postglacial accumulation of sediments in these valleys[8]. We used an artificial neural network (ANN) algorithm[11] to estimate the sediment thickness within all Alpine valleys (see 'Methods' section; Fig. 3a). Our estimates agree well with fill thicknesses observed in boreholes or estimated from seismic and gravimetric surveys (Supplementary Table 1), and they yield consistently similar or higher thicknesses where boreholes did not reach bedrock (Fig. 3c). Very shallow valley fills (<200 m) show the largest relative discrepancies, which can be attributed to distances between observation sites and valley walls that are smaller than the spatial resolution of the model (90 m). Compared with an independent sediment-

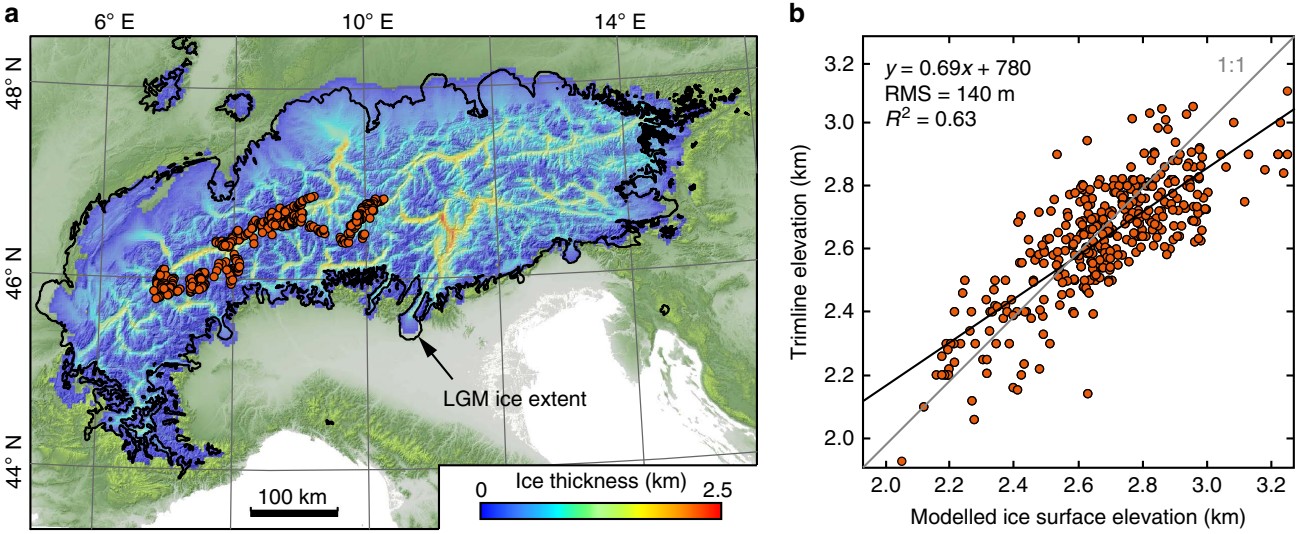

**Figure 4 | Alpine LGM ice cap.** (**a**) Steady-state ice geometry and location of trimline measurements[32–34] (orange circles). (**b**) Comparison of trimline elevations with modelled ice-surface elevations. Model resolution is 3 km.

thickness reconstruction at much higher spatial resolution (10 m) in the region around Bern[23] our reconstruction yields a fill volume of 29 km³, which is ~20% higher than the previous estimate of 24 km³, and thus reasonably consistent when considering the differences in spatial resolution. According to our estimate, the total volume of valley-filling sediments is 1,800 ± 202 km³. This volume is a minimum bound on postglacial erosion; yet it is likely only a small underestimate because a large fraction of all postglacially eroded material was trapped in a closed system of overdeepened valleys and marginal lake basins[24]. To account for dissolved fluxes and for catchments that were only temporarily closed, we added another 10% to our volume estimate[24], resulting in 1,980 ± 222 km³ of material eroded after deglaciation. Assuming mean densities of 2,000 kg m⁻³ for unconsolidated sediments and 2,700 kg m⁻³ for bedrock, we computed a total eroded mass of ~4 × 10³ Gt. Evenly distributed over the Alps (~123,000 km², excluding valley-fill areas) this volume corresponds to a rock column of 11.9 ± 1.3 m. With the assumption that the valley fills have formed following ice retreat ~17 kyr before the present (BP) (ref. 24), this eroded column corresponds to a mean postglacial denudation rate of 0.7 ± 0.08 mm yr⁻¹. To create a mass-conserving topographic basis for modelling the Alpine ice cap, we redistributed the calculated sediment volume catchment-wise and as a power-law function of local relief[25] back onto the hillslopes (Fig. 3b). Our valley fill-derived erosion rates from six lake-bordered catchments are comparable to or somewhat higher than erosion rates estimated from modern river loads[26], cosmogenic nuclides[27–29] and thermochronology[18] (Fig. 3d). Furthermore, because there might be also older sediments that predate the LGM[30] incorporated in our valley-fill estimate, our estimate of postglacial erosion and sediment redistribution is at the upper limit of probable values.

**Reconstructing the LGM ice cap.** Based on mapped ice extent and thickness indicators, such as terminal moraines[31] and trimline elevations[32–34], we reconstructed the LGM ice cover using a numerical ice-flow model[12]. Because it is not our aim to derive paleoclimatic conditions during the LGM, we used modern precipitation maps and an Alpine-wide average glacial mass-balance profile together with an iteratively adjusted equilibrium-line altitude to fit the observations (see 'Methods' section). The

steady-state LGM ice cap, which best fits available ice-extent indicators (Fig. 4) has a mean and a maximum ice thickness of 415 and 2,445 m, respectively. The maximum ice thickness is higher than previously reported values of ~2,000 m (refs 32,34) due to our removal of thick post-LGM valley fills from the underlying topography. When compared with the existing reconstruction of the LGM ice cover in the Central Alps[35] ($V = 25,000$ km³), our modelled ice volume is only ~8% larger (27,000 km³), which has negligible effects on our final results. The total reconstructed ice mass is 62 × 10³ Gt, which is ~16 times the mass of the postglacially eroded sediments.

**Lithospheric deflection.** The LGM Alpine ice cap started growing before 30 kyr BP and reached its maximum ~21 kyr BP, followed by rapid deglaciation with ~80% ice loss over the course of 3 kyr (ref. 36). This chronology is consistent with the dated onset of marginal lake formation, which indicates ice retreat to the mountain interior at 16–18 kyr BP (ref. 24). Because durations of ice-cap growth are long (>10 kyr) compared with maximum expected viscoelastic relaxation times of 3–6 kyr[6], we assume that the Alpine ice cap reached full isostatic compensation. We calculated the lithospheric equilibrium deflection[37], due to the ice loading, while accounting for a variable EET[13]. Although relative spatial variations in EET are well constrained[13], the absolute values are not, because they strongly depend on the assumed rheology and geothermal gradient. For the Alps, a range of 10–50 km has been reported in previous studies[1]. Therefore, we solve for a range of possible average EETs while maintaining the spatial pattern (see 'Methods' section). An increase of the EET results in a smaller amplitude but a larger wavelength of the deflection, which approximately follows the ice extent. With an EET of 10 km (70 km), the maximum depression is 279 m (105 m) near the centre of the ice cap and the elevated forebulge has a height of 7 m (3 m), (Supplementary Fig. 1). The effect of spatial variations of the EET on the deflection decreases with an increasing average EET. Compared with the flexural pattern that results from a constant EET of 70 km, the deflection using a variable EET of 61–80 km is up to 2 m higher in the centre, and 1.5 m lower at the periphery (Supplementary Fig. 2). Because the EET only reflects the flexural properties of the lithosphere, it does not account for deeper-seated processes, which potentially modulate the isostatic response to loading and unloading. We address these aspects in the 'Discussion' section.

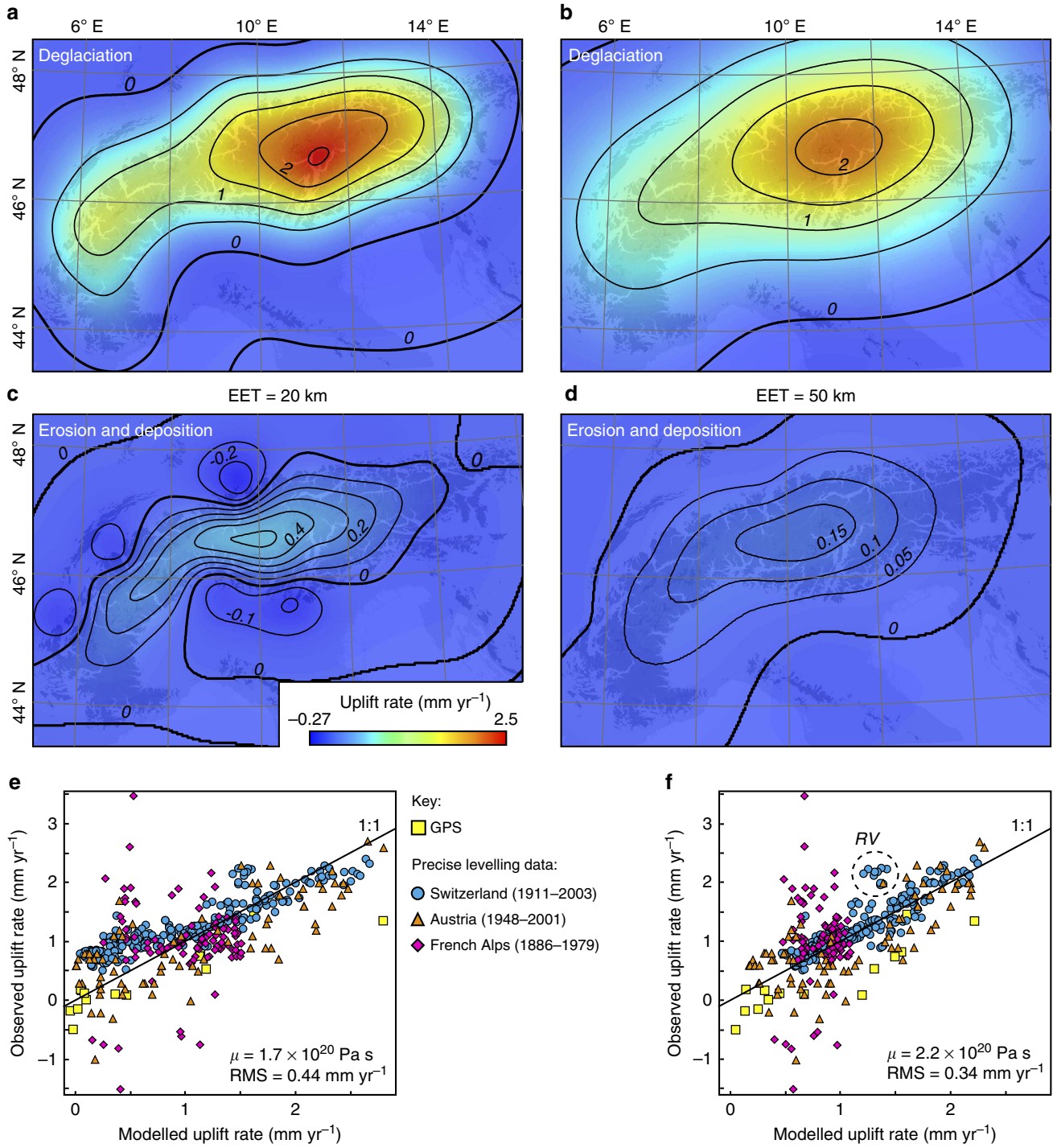

**Figure 5 | Uplift rates caused by deglaciation and postglacial erosion and deposition.** (**a**) Deglaciation component, (**c**) erosion and deposition component and (**e**) comparison of the combined signal with geodetic measurements[38–42] computed for an average EET of 20 km. (**b,d,f**), same as (**a,c,e**) but for an average EET of 50 km. Location of highlighted data cluster in **f** is shown in Fig. 6. French data were not used in the optimization (see text for details). RV, Rhône Valley.

**Isostatic rebound.** Due to the relatively small dimensions of the Alpine ice cap, the upper mantle viscosity determines isostatic uplift and relaxation rates. For a range of mantle viscosities of $10^{19}$–$10^{21}$ Pa s, we calculated the corresponding uplift rates associated with deglaciation, which we assume to have occurred at $17 \pm 2$ kyr BP (see above paragraph on postglacial erosion). To account for the isostatic component related to erosional unloading, we calculated the lithospheric response to the estimated erosion/deposition for average EETs of 10–70 km,

assuming that the erosional mass redistribution is compensated by the mantle at a steady rate. We finally compared the combined uplift signal of GIA and erosional unloading with measurements obtained from precise levelling[38–40], which were adjusted to a common reference frame using data from permanent GPS stations[41] (see 'Methods' section).

With an average EET of 50 km and an upper mantle viscosity of $2.2 \pm 0.5 \times 10^{20}$ Pa s, we are able to reproduce virtually all of the geodetically measured uplift with ∼90% of this uplift

caused by deglaciation and only ∼10% by erosional unloading (Fig. 5f). Comparison with independent measurements from the French Alps[42] shows that 70% of the observed uplift rates are near the model-predicted values (Fig. 5f). The derived viscosity is reasonable, as it is lower than the viscosity estimated for an old craton (3–10 × $10^{20}$ Pa s, Fennoscandia), but higher than that for a region with recent crustal thinning (0.18 × $10^{20}$ Pa s, Basin and Range province, Supplementary Table 2). For illustrative purposes, we repeated our calculation assuming that 90% of the postglacially eroded material has been exported from the Alps. Similar to our previous results, the best fit with the data is achieved with an EET of 50 km and an upper mantle viscosity of 1.8 × $10^{20}$ Pa s, whereas the erosional contribution to the total uplift rates increases to a maximum of 35% (Supplementary Fig. 3).

## Discussion

It has been argued that the geodetically measured uplift of the Alps is dominated by its isostatic response to erosional unloading[8], which is supposed to have increased threefold from Pliocene to Quaternary times[43], although some of this increase may be an artefact of incomplete preservation of older deposits[44,45]. Several thermochronometric studies point to an increase in exhumation rates of similar magnitude over the last 2 Ma, which has been attributed to a positive feedback between glacial incision, isostatic rebound, rock uplift and exhumation rates[18–20]. However, we argue that much of the material eroded from the Alps since deglaciation was deposited within Alpine valleys and therefore does not contribute to erosional unloading. Our models show that the LGM ice load (∼62 × $10^3$ Gt) was much larger than postglacially eroded sediments (∼4 × $10^3$ Gt) and suggest the dominance of ice melting over erosional unloading in contributing to the total recent uplift rate. Furthermore, erosion rate estimates based on our valley-fill volumes exceed long-term rates based on thermochronology (Fig. 3d). This could be explained by a peak in erosion rates at the onset of deglaciation due to intensified paraglacial processes that can be expected for a landscape that is adjusting to new boundary conditions. Even in the unreasonable case that 90% of the postglacially eroded material had been exported from the orogen, the erosional contribution to the recent uplift rate would not exceed 35%.

The most sensitive parameter in our modelling that is not well constrained is the average EET. A low average EET does not change the best-fit viscosity by much (1.7–2.2 × $10^{20}$ Pa s), but leads to systematically lower modelled uplift rates in the northwestern periphery of the Alps (Fig. 6a). Active tectonic shortening across this region is <1 mm yr$^{-1}$ and contributes <0.2 mm yr$^{-1}$ of rock uplift[8]. If other processes, such as lithospheric delamination[10] or ongoing tectonic shortening[46] were to account for some of the observed uplift in the northwest periphery of the Alps, these would have to generate the same uplift pattern as the ice unloading, which we think is rather unlikely. We therefore favour our model results with a relatively thick EET (50 km), in which ∼90% of the present-day uplift of the Alps is due to GIA. However, we acknowledge that the Alpine lithosphere has a complex architecture[47] that our elastic thin-plate approach may not be able to fully account for.

For example, we observe a conspicuous cluster of residual uplift in the Swiss Rhône Valley, which is close to a zone of enhanced seismicity and may thus have a tectonic origin (Supplementary Fig. 4). Recent analysis of geodetic data in the Western Alps has revealed a narrow zone of particularly high uplift rates (1.5–2.5 mm yr$^{-1}$) that exceed the combined signal of GIA and erosional unloading by up to 1.5 mm yr$^{-1}$ (ref. 22) (Supplementary Fig. 4b). This excess uplift may be attributed to a contrast in crustal viscosities between the foreland and the

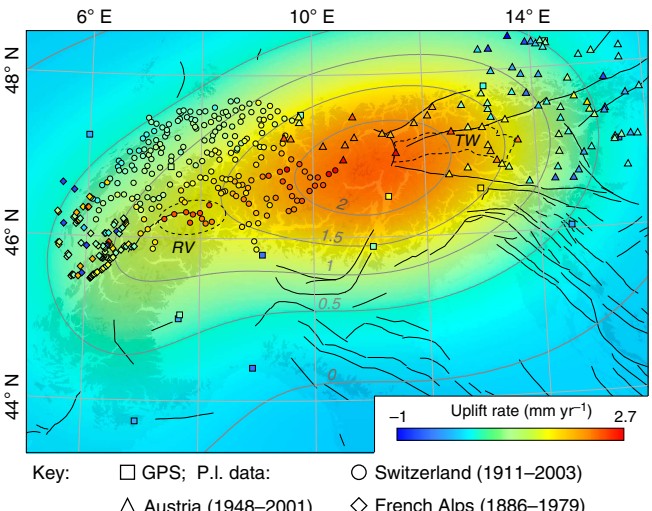

**Figure 6 | Observed and modelled uplift rates.** Modelled uplift rate for an average EET of 50 km with the adjusted geodetic measurements[38–42] superimposed. Black solid lines indicate seismogenic faults (http://diss.rm.ingv.it/share-edsf/). Dashed ellipse indicates cluster of excess uplift in the Rhône Valley (RV). TW, Tauern Window (dashed outline).

mountains, which could enhance the isostatic response to crustal unloading[48]. Because the zone of high uplift coincides with a low-P-wave velocity anomaly at 100–150 km depth[10] and with high Quaternary exhumation rates[18], it may also (or alternatively) be related to deep-seated mantle processes that act on million-year timescales[22]. We thus speculate that the excess rock uplift we observe in the Swiss Rhône Valley may be linked to crustal heterogeneity, or deep-seated processes.

Because of the ongoing N–S convergence, we expect a larger tectonic contribution to the recent uplift rates in the Eastern Alps. Estimated rock uplift rates in the Tauern Window (Fig. 2) and along the eastern edge of the Alps since 4 Ma are <1 mm yr$^{-1}$ (ref. 9) and 0.1–0.15 mm yr$^{-1}$ (ref. 49), respectively, and thus one order of magnitude lower than geodetically observed uplift rates. Adding this tectonic component to our modelled uplift rate would in both cases improve the fit with the measurements. Therefore, the mismatch between modelled and observed uplift rates in parts of Austria could be explained by the ongoing convergence in the Eastern Alps that is related to the counterclockwise rotation of the Adriatic microplate, and by the eastward extrusion of the Eastern Alps that is associated with a complex pattern of transtensional and transpressional zones[15,21](Fig. 2). We thus conclude that the recent long-wavelength uplift of the European Alps is predominantly due to GIA with a small erosional contribution of probably <10% and that observed residuals are likely due to local tectonic activity and deep-seated mantle processes.

Although the Alpine LGM ice cap was much smaller than the northern hemisphere ice sheets, the Earth's response to its demise dominates the vertical motion over broad regions of the Alps even today. Other mountain belts that were similarly affected by glaciation, such as the Alaska Range, the Himalaya or the Southern Alps in New Zealand, might therefore also exhibit such a long-wavelength uplift component.

## Methods

**Estimation of valley-fill thicknesses.** Based on the assumption of geometric similarity between the exposed and the buried parts of the landscape, we used an ANN algorithm[11] and a 90- m-resolution digital elevation model (DEM) to explicitly estimate the depth to bedrock for grid cells that include valley fill. We expect geometric similarity of the bedrock surface, because the entire landscape was

subject to glacial erosion before deposition of the valley fill. For each cell in the DEM that is part of a valley fill, the depth to bedrock is estimated from the horizontal distance to the nearest hillslope, calculated for different directions. Training and validation of the ANN was initially performed on the presently exposed topography using synthetic fills. We created a mask of all valley fills by a combination of slope thresholds and manual digitization using geological maps with a scale of 1:25,000 (https://map.geo.admin.ch) and 1:50,000 (https://www.geologie.ac.at, http://infoterre.brgm.fr, http://www.geoviewer.isprambiente.it). To calculate the eroded rock column for each cell since the LGM, we adopted the proposed relation between erosion rate ($E$) and mean local relief ($R$, determined in a window with 10 km diameter) of $E = 1.4 \times 10^{-6} R^{1.8}$ (ref. 25) and multiplied this by 17,000 years. We then calculated the fractional contribution of each cell by multiplying with the rock volume derived from the valley-fill approach.

**Ice-cap reconstruction.** To reconstruct the Alpine ice cap during the LGM, we employed a numerical ice-flow model[12] to solve the shallow ice approximation (SIA)[50]. The SIA simplifies ice dynamics but allows for computational efficiency at sufficiently high spatial resolution (3 km). We acknowledge that the SIA-approach may not be suitable concerning the calculation of the ice velocity and flow patterns in steep terrain. GIA, however, is ultimately controlled by the ice geometry and is insensitive to the ice velocity and flow pattern. Compared with higher-order models, only slight differences in glacier geometry are expected during steady-state[51]. The ice rheology is governed by Glen's flow law, $\varepsilon_{ij} = A\tau_e^2 \tau_{ij}$, where $\varepsilon_{ij}$ are the components of the strain rate tensor, $\tau_{ij}$ are the components of the deviatoric stress tensor, $\tau_e$ is the effective stress and $A = 1 \times 10^{-16}\,\mathrm{Pa}^{-3}\,\mathrm{yr}^{-1}$. The sliding velocity $u_s$ is assumed to be proportional to the basal shear stress $\tau_s$, and given by: $u_s = A_s\tau_s^2 N^{-1}$. $A_s$ is a sliding coefficient depending inversely on the bed roughness and $N$ is the effective pressure at the base of the ice and set to 40% of the ice overburden pressure. In reality $N$ would be highly variable in both space and time[52] with lower values leading to increased decoupling between ice and bed, which permits faster sliding. Tests with $N$ being 80% of the ice overburden pressure indicate only a small increase in the resulting ice volume. The surface mass balance is modelled with an accumulation/ablation gradient of 0.7 m snow water equivalent per year $(100\,\mathrm{m})^{-1}$ (ref. 53) and a spatially variable maximum accumulation rate, using the recent pattern of mean annual precipitation[54] (Supplementary Fig. 5). To achieve the best match between modelled and mapped ice extent and thickness, we iteratively adjusted equilibrium line altitudes for each catchment that drains the Alps according to the areal misfit determined after each model run (Supplementary Fig. 6). This adjustment was repeated for $A_s = 25, 75, 100, 150$ and $200 \times 10^{-10}\,\mathrm{m\,yr^{-1}\,Pa^{-2}}$. The best agreement between mapped trimline elevations[32–34] and the modelled ice-surface was reached with $A_s = 100 \times 10^{-10}\,\mathrm{m\,yr^{-1}\,Pa^{-2}}$. Furthermore, we increased $A_s$ stepwise within the foreland to prevent the Alpine ice cap from overtopping the Jura Mountains. Field observations clearly document that two branches of ice were flowing to the northeast and to the southwest of the Jura Mountains[55]. This pattern was reproduced when $A_s$ in the foreland was increased by a factor of 15, which is similar to the value used in a reconstruction of the Laurentide ice sheet[56]. We attribute differences in the sliding coefficient between the foreland and mountain interior to deformable sediments[57] and higher amounts of meltwater[52], which both are likely associated with higher sliding velocities.

**Flexure of the lithosphere.** Because the lithospheric deflection due to glacier growth has a direct effect on the slope and elevation of the ice surface and hence on the ice flow and mass balance, we reconstructed the ice cap for uniform EETs of 20, 30, 40 and 50 km, respectively. We calculated the flexural isostatic adjustment, $W_f(x,y)$, for every 10 model time steps ($\sim$30 days) using the two-dimensional elastic thin-plate equation:

$$\frac{\partial^4 W_f}{\partial x^4} + 2\frac{\partial^4 W_f}{\partial x^2 \partial y^2} + \frac{\partial^4 W_f}{\partial y^4} = \frac{L(x,y)}{D_f} \tag{1}$$

Here $D_f = Y\mathrm{EET}^3/12(1-v)$ denotes the flexural rigidity, where EET is the effective elastic thickness of the lithosphere, $Y = 100\,\mathrm{GPa}$ is Young's modulus and $v = 0.25$ is the Poisson ratio. $L(x,y) = \rho_i g H(x,y) - \rho_a g W_f(x,y)$ is the vertical load where $\rho_i = 917\,\mathrm{kg\,m^{-3}}$ is the density of ice, $H(x,y)$ is the ice thickness in each model cell and $\rho_a = 3,300\,\mathrm{kg\,m^{-3}}$ is the density of the compensating asthenosphere. The variation in total ice volume and maximum ice thickness due to an increase of the EET from 20 to 50 km is $<5\%$. To further investigate the effect of a laterally heterogeneous lithosphere, we introduced variations in EET[13], and calculated the isostatic depression due to the load of the steady-state ice cap using the algorithm gFlex[37], which uses finite difference solutions for the problem of elastic plate bending under arbitrarily shaped surface loads. To account for the effect of the differences in the resulting flexural patterns on the ice geometry, we adjusted the glacier bed according to the results from gFlex and continued running the ice model to find the new steady-state ice geometry (Fig. 4). Imposing a variable EET results in $1 \times 10^3\,\mathrm{km^3}$ more ice, which is small when compared with the total ice volume of $68 \times 10^3\,\mathrm{km^3}$. The larger ice volume can be attributed to thicker ice (up to 160 m) in the foreland lobes.

**Effective elastic thickness.** EET of the lithosphere exerts a primary control on the plate's flexural rigidity, which in turn determines the magnitude and pattern of the isostatic response. Reflecting the long-term and often complex history of the continental plate, the EET depends mostly on the combined effects of rheological and thermal heterogeneity. In this study we use EET estimates[13] obtained by following the approach of ref. 58. The crustal rheology, corresponding to quartzite in the upper crust and diorite in the lower crust[59], was assigned using the velocity distribution of the crustal model EuCRUST-07, which is based on integration of several hundred seismic profiles and receiver-function data[60]. For the mantle lithosphere, a 'dry' olivine rheology was used. Lithospheric temperatures were derived from the inversion of a tomographic model of seismic velocities for Europe[61]. The EET ranges from 14 to 27 km in the Alpine region, with overall higher values in the Eastern Alps compared with the Central and Western Alps (Supplementary Fig. 1a). However, these values reflect only a lower bound endmember, assuming a high geothermal gradient and a 'soft' rheology ('dry' quartzite and 'wet' diorite)[13]. Therefore, we modified the corresponding EET by adding 10, 20, 30, 40 and 50 km to the absolute values to account for lower geothermal gradients and a stiffer rheology. Larger values of EET than 50–70 km imply a significant contribution of the upper mantle to the rigidity of the lithosphere and are likely more representative of the Precambrian cratons than of active Phanerozoic regions like the Alps[59].

**Rebound model.** The viscoelastic decay of the lithospheric deflection after removal of the surface load results in uplift at a rate $u$, which we calculated with an exponential decay model:

$$u = -W_0/\tau \times \exp(-t/\tau) \tag{2}$$

where $W_0$ is the equilibrium deflection, $\tau$ is a characteristic timescale of relaxation and $t$ is the time since unloading[62]. The timescale of relaxation is defined as:

$$\tau = 4\pi\mu/\rho g\lambda \tag{3}$$

where $\mu$ is the mantle viscosity, $\rho = 3,300\,\mathrm{kg\,m^{-3}}$ is the mantle density, $g$ is the gravitational acceleration and $\lambda = 320\,\mathrm{km}$ is the wavelength of the load, which we calculated from the average transverse extent of the LGM ice cap. We used uplift rates determined by precise levelling in Switzerland[38,39] and Austria[40], obtained from repeated measurements of benchmarks since the beginning and the middle of the 20th century, respectively. Thus, they represent vertical velocities in relation to arbitrarily chosen reference points. The reference point for the Swiss data is located near the city of Aarburg on the Swiss Plateau and the Austrian data refer to a point near the city of Horn $\sim$70 km northwest of Vienna (Fig. 3a). Direct comparison of both datasets requires that the vertical velocities of the respective reference points can be determined. For this purpose, we used data from the permanent GPS stations Zimmerwald (Z) and Linz (L), which are located on the Swiss Plateau and the Bohemian Massif, respectively. Both stations provide continuous time series of ground motion from 1998 to 2015 (17 yr) and from 2005 to 2013 (8 yr), with vertical velocities of $1 \pm 0.08\,\mathrm{mm\,yr^{-1}}$ (Z) and $0.8 \pm 0.15\,\mathrm{mm\,yr^{-1}}$ (L) in the global reference frame IGb08 (ref. 41). To adjust the measurements to a common reference frame we subtracted the GPS rates with the uplift rates of the nearest levelling benchmarks, which resulted in uplift of $0.94 \pm 0.08\,\mathrm{mm\,yr^{-1}}$ for the Swiss reference point and $1 \pm 0.15\,\mathrm{mm\,yr^{-1}}$ for the Austrian reference point. We inverted the adjusted uplift rates for the mantle viscosity using equation (2) with $t = 17 \pm 2\,\mathrm{kyr}$, resulting in a viscosity of $1.4–2.8 \pm 0.5 \times 10^{20}\,\mathrm{Pa\,s}$, depending on the assumed EET (20–50 km), with larger EETs leading to higher viscosities (Supplementary Fig. 7).

**Code availability.** gFlex is available from Andrew Wickert's GitHub repository at https://github.com/awickert/gFlex.

**Data availability.** The data that support the findings of this study are available from the corresponding author upon request.

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

## Acknowledgements

This research was funded by the Potsdam Research Cluster for Georisk Analysis, Environmental Change and Sustainability (PROGRESS) through a grant of the German Federal Ministry for Education and Research (BMBF) to M.R.S., J.M. and

T.F.S. acknowledge funding by the Deutsche Forschungsgemeinschaft (DFG) through Emmy Noether grant SCHI 1241/1-1 awarded to T.F.S., M.T. acknowledges Utrecht University and the Netherlands Research Centre for Integrated Solid Earth Science (ISES) for funding through grants ISES-2014-UU-08 and ISES-2016-UU-19. This work benefited from discussions with J.P. Avouac, J. Braun, M. Handy, P. van der Beek and F. von Blanckenburg. We acknowledge constructive comments by Fritz Schlunegger and two anonymous reviewers.

## Author contributions

D.S. and J.M. conceived and designed the study with the help of A.D.W., M.R.S. and T.F.S. J.M. carried out the modelling work and wrote the paper together with D.S. A.D.W. helped computing lithospheric flexure. D.L.E. helped with the ice model and M.T. provided the map of the elastic thickness of the Alpine lithosphere. All authors contributed to the discussions, interpretations and writing of the manuscript.

## Additional information

**Competing financial interests:** The authors declare no competing financial interests.

DOI: 10.1038/ncomms16138   **OPEN**

# Corrigendum: Glacial isostatic uplift of the European Alps

Jürgen Mey, Dirk Scherler, Andrew D. Wickert, David L. Egholm, Magdala Tesauro, Taylor F. Schildgen & Manfred R. Strecker

*Nature Communications* 7:13382 doi: 10.1038/ncomms13382 (2016); Published 10 Nov 2016; Updated 14 Jul 2017

In an earlier publication, Norton and Hampel proposed post-glacial uplift promoted the re-advance of glaciers at the onset of the Younger Dryas by enlarging their accumulation areas, and estimated a maximum present-day uplift rate due to deglaciation of $\sim 0.36$ mm per year, approximately six times smaller than the value presented in this Article (2.3 mm per year). We suggest this discrepancy is a result of different model assumptions regarding the structure and rheology of the lithosphere, the ice mass and the unloading history.

While this publication was initially omitted from the reference list of this Article, the authors acknowledge that given the differences in the studies' conclusions, citation of this earlier work is wholly appropriate.

Norton, K. P. & Hampel, A. Postglacial rebound promotes glacial re-advances—a case study from the European Alps. *Terra Nova*, **22**, 297–302 (2010).

