## [Peer Review File · Nature Communications]

Reviewer #1 (Remarks to the Author):

Review of "Glacial isostatic uplift in the European Alps" by Mey et al.

In this paper, the authors propose that the current uplift of the Alps is due to viscoelastic response of ice unloading since the last glacial maximum. To do this, they use a reconstruction of the glacial overdeepening, an ice flow model and flexural model to explain the available levelling data. The paper is nicely written, clear and includes thorough calculations. I think it may eventually be published in Nature Communications. However, I have several concerns that I think should be addressed before recommending publication.

First and foremost, I am really concerned about the authority used to express the conclusions that all the uplift must solely be explained by the Earth's viscoelastic response to ice unloading, at the expenses of erosion or tectonics. The authors' analysis is interesting, but it does include a range of poorly constrained parameters. More importantly, the study does not disprove the proposition that about 60% of the uplift can be explained with a flexural model, and all of it with local isostasy only (Champagnac et al., 2009). The Alps are currently eroding (e.g., Hinder, Champagnac, Wittmann, to cite a few), so there must be a response. How much is debatable, but arguing that there is no uplift due to erosion is equivalent to saying that flexural (and to some extent local) isostasy does not work. Maybe I am misunderstanding, for example about how the erosion correction that is done, but the abstract statement is very strong: "we show that virtually all of the geodetically measured surface uplift in the European Alps can be explained by the Earth's viscoelastic response to ice unloading after the last glacial maximum (LGM)". So if the authors want to keep this stand, they should demonstrate convincingly that erosion does not cause any uplift.

Second, the use of ice flow model should be better justified. There exists an ice reconstruction (Bini et al., 2009) that could be used, so I did not fully understand why the authors needed to run models, especially that the climate part is poorly constrained. Furthermore, the modelled ice volume is different than the reconstructed one. That must affect the calculations. A comparison of modelled rock uplift rates between the icecap reconstruction and the modelled icecap would be useful to assess whether it matters. It would definitely justify the model better.

Third, the final discussion is very disappointing. The authors talk about specific aspects about the Alps. Instead, or in addition, they should discuss the implications of their claims to our understanding of mountain ranges in general.

Line by line comments:

Line 47: change 'furnish' by 'provide'

Line 57-60: It would only be fair to mention some of the recent papers that gave constraints on the timing of increase erosion in the Alps, as it is given for the other aspects of the Alps geological history. There were several papers in recent years (e.g. Veron, Glotzbach, Valla, Fox to cite a few).

Line 75-76: using 'surface lowering' is confusing here. It seems it does not include isostatic response to erosion. Please clarify this statement.

Line 87-88: Please explain better why there is such a large difference in ice thickness. What is correct, the reconstruction or the ice thickness? What is the effect of such difference on the

rebound?

Line 125: Same comments as for line 57-60. There are thermochronometric studies do not suffer for the incomplete preservation (though they suffer from other problems...).

Line 126-127: Point well taken, but that does not disprove the alternative option that erosion causes the uplift. It is not because your model fits the data that it disproves the other option.

Line 130-144: See main comment.

Line 188: remove "the" before Poisson

Line 241: change "inverted" by "optimized"

Reviewer #2 (Remarks to the Author):

The authors of this paper claim that the current uplift of the European Alps, measured with geodetic techniques, is the response to glacial unloading since the termination of the LGM c. 17 Ka ago. They used a 2D elastic thin-plate model overlying a viscous mantle as a proxy for modelling the response of the European lithosphere to unloading, and an inferred thickness pattern of LGM ice derived through the use of a numerical ice flow model to solve the shallow ice approximation. The authors also performed a sediment budget to account for the contribution of Holocene erosional unloading to the currently measured uplift. The authors indeed address a problem that has not been solved during the past decades, and as such any improvements in this field will be of great significance. I am much in favor of seeing this topic addressed, discussed and results being published, particularly in this journal that is designed for a broad readership, but I do see space for improving the current manuscript. The major points include: (i) a better explanation of the novelty of this research, (ii) a better justification of the selected lithospheric model and ice flow model, (iii) a more careful treatment of how published data is being handled, and (iv) a more careful consideration of the data and processes in the Eastern Alps.

Let me first outline my first point, and this addresses the novelty of the presented manuscript.

Gudmundsson (1994, paper cited in this paper) addressed the same question for the Central Alps, used a similar approach and came to the same conclusions, both regarding the glacial unloading as a major driving force, the inferred viscosity of the underlying mantle, and the deviations between model and observation particularly in the Valais. In the same sense, also applying a similar modeling approach, Champagnac et al. (2009, cited in this manuscript) came to different conclusions in the sense that most of the uplift we currently see in the Central Alps is due to erosional unloading.

Champagnac et al. also detected deviations between geodetic uplift and their model particularly in south, similar to the here presented study. The authors of this manuscript mention this and say that the LGM ice load was much larger than the post-glacially eroded sediments. They then refer to their own model that suggests the dominance of ice melting over erosional unloading. I have not been satisfied by this explanation as it does not clarify the situation. This then means that the differences between the conclusions by Champagnac et al. and this paper remain non-discussed. In the same sense, it is not clear to me how much of the geodetic uplift is still related to the geodynamic processes operating at much larger time scales (e.g., Baran et al., 2014, *Lithosphere*; Fox et al., 2015, *Geology*; Wagner et al., 2010, *EPSL*). For instance, how much of the uplift measured in the Eastern Alps is due to the rotational collision of the Adriatic plate? I do acknowledge that the authors focus

their analyses on the Holocene and the modern situation, but some of the modern uplift (if not all?) could also be related to long-term processes including slab breakoff (beneath the Western Alps) and intra-crustal delimitation (papers cited above), and clockwise rotation of the Adriatic plate with the potential to shorten the Eastern Alps. If the authors have sorted out these questions and still conclude that most of the modern uplift is due to glacial unloading, then this paper would represent a breakthrough in our understanding of the Alps. In summary, I see the need that the authors: (i) explain the differences between their work and the results by Gudmundsson (1994), (ii) properly discuss why their conclusions should be more valid than those by e.g., Champagnac et al. (2009), and (iii) disentangle between the long-term and short-term controls/mechanisms of uplift.

My second concern addresses the selection of the models. I do appreciate that thin-plate approaches have been conventionally been used for modeling the deflection of the European continental plate, and I admit that I have done the same before. I am still convinced that this is a valid approach for the Scandinavian lithosphere that extends over thousands of kilometers. However, collisional orogens such as the Alps have a much different lithospheric architecture that comprises (for the central Alps): (i) the c. 1.5 km-thick topographic load made up of upper crustal rocks, (ii) the c. 50 km thick crustal root made of upper and lower crustal material; the crustal root is buoyant and explains most of the negative Bouguer gravity anomalies of c. 180 mGal in the Central Alps (Lyon-Caen Molnar, 1989; Holliger & Kissling, 1992), plus (iii) the subducted c. 160 km-long lithospheric mantle that yields a downward directed slab load force (Lippitsch et al., 2003, also cited in the manuscript). Accordingly, while a c. 50 km thick lithosphere that extends over thousands of kilometers can be modeled with confidence by the thin plate approach, the dimensions in the Alps (>50 km thick crustal buoyant root beneath the core of the Alps plus the 160 km-long subducted lithospheric mantle) makes this justification more difficult. Besides, the slab hangs nearly vertically beneath the Bavarian Alps (Lippitsch et al., 20013), suggesting nearly zero mechanical strength of the plate, at least in this part of the Alps. Contrariwise, the slab dips with only 60 degrees beneath the Central Alps. This documents the occurrence of some remaining flexural rigidity in the European lithosphere. Beneath the Western Alps, the slab has possibly broken off (Lippitsch et al., 2003). This yields a complex 3D architecture that most likely imprints on the most recent uplift and exhumation history (Baran et al., 2014; Fox et al., 2015). The authors account for spatial variations of the elastic properties of their model plate, without however explicitly referring to the results of geophysical and seismo-tomography surveys. My point is that the deep structure beneath the orogen should not be discarded (Moho, crustal root, subducted or broken slabs etc.) and needs to be more carefully discussed. We have presented a simplified sketch of this in Schlunegger and Kissling (2015), based on which either the simplest case of Airy-Heiskanen isostasy could be applied where the local accommodation of crustal thicknesses conditions the topographic heights. Alternatively the more complex case of Vening Meinesz isostasy where loads are regionally compensated might be more appropriate. I do acknowledge that I have a biased view on this since I have worked with these models recently. But whatever approach is the most appropriate for collisional orogens: The authors might be correct in selecting their preferred model, but this warrants a better justification particularly considering the architectural complexity of collisional orogens.

In the same sense, the selection of the shallow ice approximation for reconstructing the ice cap needs to be better justified. Related equations have been derived for thin ice, where the ice surface and the bedrock topography at the base of a glacier are nearly parallel to each other (approach selected in this paper). In this case, the basal shear stress τ_b has been related to the ice thickness and

the gradient of the ice surface. However, this model does not consider processes such as dislocation creep and recrystallization processes within the ice, driven by pressure gradients of the overlying ice. These mechanisms turn thick ice into a viscous material and could play an important role particularly when explaining the velocity fields of thick trunk glaciers, and I guess that this was the case during the LGM. The authors justify their simplifications because it allows them to compute more efficiently the buildup of the ice cap, which is a fair compromise. However, one problem with the thin ice approach still remains, and this lies in the misfit between the ice surface and bedrock topography particularly when comparing the Bini et al. (2009) map and the current landscape of the Central Alps. The two surfaces are not sub-parallel to each other, particularly for regions surrounding current lakes. Finally, the use of recent annual precipitation does not account for the inferred S-derived moisture sources during the LGM, as proposed by Florineth and Schlüchter (2000). In summary, I propose the author to better explain the preferred selection of their ice growth model along why the current precipitation pattern serves as a proxy for the LGM times, although other authors proposed that atmospheric circulations were different in the past (e.g., Florineth and Schlüchter, 2000).

As the third point: I am wondering why the authors have not taken the ice cover map by Bini et al. (2009) to constrain their LGM ice model, at least for the Central Alps. The map has been reconstructed based on detailed observations of trim-lines in the Central Alps and considering available local maps. It is available from Swisstopo upon request and can be easily transferred into an ice thickness map. In this context, I am wondering why the rebound effect should not be much larger in the foreland given the relatively thick LGM ice there (e.g., nearly 1000 m-thick LGM ice above lake Geneva, and c. 700-800 m above Lake Constance). In either case, I strongly recommend using the Bini et al. (2009) LGM reconstruction to constrain ice growth models, else justifications/explanations are needed.

In the same sense, I have been wondering why the authors have not used existing bedrock topography models to constrain their estimation of valley fill thicknesses. In particular, there exists a bedrock topography map compiled for the Swiss Alps by Dürst Stucki and Schlunegger (EPSL); the data is also available upon request from Swisstopo. This map is based on a compilation of thousands of drillings and has a yet unbeatable resolution, at least to my perception. Last but not least, the statement that the valley fill must have formed following ice retreat at c. 17 Ka (line 76) is not correct as the valley fill sediments are older than the Holocene. For instance, some of the overdeepening fills have been dated around MIS8, or 270 Ka (Preusser et al., 2010; 2011; Dehnert et al., 2012). Older ages for the Thalgut site of 420 Ka have been reported by Preusser et al. (2011). Therefore the erosional budget presented here needs to be corrected. Since the authors claim that the contribution of erosional unloading is minor, the correction will not alter the main conclusion. Finally, one needs also to consider that valley fills do not account for the solute loads, which might be up to 50% and even more in some cases (see articles by Hinderer and colleagues). This is the major reason why the ^{10}Be -based basin averaged erosion rates might be more suitable for estimating the amount of erosion unloading at least during the Holocene (Champagnac et al., 2009) as this methodology includes unloading through physical and chemical erosion (Wittmann et al., 2007).

As a fourth point, not much is said about the Eastern Alps and the Western Alps (this actually also concerns my review, which has a strong bias towards the Central Alps). The authors claim to present

a view across the entire Alps, but the discussion mainly focuses on the central portion. For instance, Wagner et al. (2011) thought that effects from the counterclockwise rotation of Adria could be seen by the young uplift of the Styrian Basin in the eastern Alps. This rotation might have resulted in crustal shortening and underthrusting of the Pannonian fragment. There is quite a long list of papers that have discussed the most recent tectonic processes in the Eastern Alps the authors need consider (Wagner et al., 2010; 2011; Wölfler et al., 2012, and others).

In summary, this has the potential to be great and heavily cited paper addressing a topic that currently receives much attention, but existing literature need to be better incorporated, selected models better and more carefully justified, the eastern and western Alps need more attention, and long-term geodynamic controls on uplift needs to be disentangled from short-term effects.

Kind regards

Fritz Schlunegger

Bern, 27th of March 2016

Reviewer #3 (Remarks to the Author):

Overall, I thought this was a very strong paper and have no major qualms with its publication as is. It is clear that a lot of work has gone not only into the research in the paper but also into the writing of the paper, as no typos or other small errors were noticeable. Overall, this is a very well written paper that successfully documents and describes the interesting study that the authors have undertaken.

Here are my organized comments based on the organized points:

A. Summary of the key results

Previous work on the Alps has generally been inconclusive on the specific cause of the modern patterns of geodetically measured surface uplift or has often attributed the uplift to the redistribution of sediment due to glacial erosion and deposition. Using a combination of paleoglacial modeling, viscoelastic modeling of the lithosphere and upper mantle, and redistributing the valley fills back up to higher elevation, this current study seeks to determine the dominant process in creating the modern and observed surface uplift. By varying properties within realistic or specific ranges, such as the elastic properties of the lithosphere and the glacial sliding coefficient, the authors were able to reasonably match modern measured uplift, valley fill thickness, and glacial trimline data. From this, they concluded that, for the most part, measured uplift in the Alps can mainly be attributed to isostatic rebound due to the lost off icecaps since the LGM, with local strong tectonic components in some isolated regions.

B. Originality and interest

This research, while building off of previous work surrounding uplift in the Alps, certainly stands on its own as a new approach to better understanding the interplay of different processes in this and similar mountain ranges. This work should be of interest not only to those who particularly interested in the Alps but also researchers who generally study sediment sources related to glacial erosion and landscape formation.

C. Data & methodology

In general, this is a novel approach to understanding uplift in the Alps. Only recently, has there been the ability to effectively combine several types of models (for example, lithospheric flexure and glacial flow) along with topographic/sedimentary redistribution. This research certainly takes advantage of the state-of-the-art techniques in these fields, while also exploiting the existing measurements of sediment depth and high precision geodetics.

As for the glacier model, there are some minor criticisms that likely only come to mind due to the length constraints of a Nature Communication paper. While the EET and lithospheric/mantle rheology variables are chosen based not only on reducing the uplift residuals, and upper mantle viscosity is compared realistically to other estimates from different regions, the variables used in the glacial model seem less explicitly informed by reality. This is not too say that they are poorly chosen but just that little justification or comparison is given for variables such as the choice of overburden pressure or the sliding coefficient. In fact the definition of the sliding coefficient, "depending inversely on the bed roughness," makes it seem like this term could in fact be related to the existing topography. While they attribute the foreland and mountain interior differences in this term to the existence or lack of deformable sediments and more meltwater, comparing the actual values and the choice of increase amount to other model results could at least ensure that the glacial modeling is consistent with previous uses of this and similar models.

D. Appropriate use of statistics and treatment of uncertainties

The only significant statistics in the paper relate to the model fitting, particularly the use of RMS. This is an entirely reasonable and appropriate method for the analysis as shown.

E. Conclusions

By showing the sensitivity of the model in Extended Data Figure 5, the author's emphasize the robustness of their results for their given models. The conclusions stand strongly against the observations. Even the regions of higher RMS are systematic and make sense given the tectonic situation of the Alps.

F. Suggested improvements

The additional information, expanded upon in Section C, that might be beneficial would be a comparison or table (similar to Extended Data Table 2) that compares the glacier flow parameters to those used in other models.

G. References

References seem entirely appropriate and nothing else seems necessary.

H. Clarity and context

The abstract is fairly straightforward and properly summarizes the paper. The only suggestion might be to end the abstract on the overall idea of glacial rebound being the dominant mechanism instead of ending on the exceptions (Swiss Rhone Valley, Eastern Alps). All other sections of the paper are clear and appropriate.

We greatly appreciate the constructive criticism offered by the reviewers, which have led us to make several important modifications and improvements to our manuscript. In the following, we give an overview on the two main points that were raised by the reviewers and how we have taken these into account, followed by a point-by-point response to each of the reviewers concerns.

1. Glacial rebound vs. erosional unloading.

Reviewers 1 & 2 criticize that we only briefly dealt with the uplift component that is due to erosional unloading. They asked us how we reconcile our finding with the work of Wittman et al. (2007) and Champagnac et al. (2009), who argue that the present-day uplift is dominated by the isostatic response to erosional unloading. To better approach these questions and address the reviewers' concerns, we first calculated postglacial erosion rates based on our valley-fill volume estimate and an additional 10% to account for material that has been exported (clastic and dissolved). We compared these rates with data derived from modern river loads, cosmogenic nuclide-derived denudation rates and thermochronology, and we found that the rates derived from the valley fills are comparable to the other estimates but rather on the higher end of the reported values. Furthermore, our inferred postglacial erosion rate for the entire Alps is 0.7 mm/yr, which is similar to the value of 0.64 mm/yr proposed by Hinderer (2001). Thus, we argue that our estimate of the postglacially eroded sediment volume is reasonable, and if anything, then rather at the upper end of probable values. When we calculated the lithospheric response to the combined deglaciation and postglacial erosion/deposition rate (assuming 10% of the material was exported and the rest was trapped), we found that erosional unloading contributes to only 10% of the recent uplift rates. To investigate how this result is influenced by the assumed relative proportion of trapped and exported material, we repeated our calculation for the extreme (we consider unreasonable) case that 90% of the postglacially eroded material has been exported. Even in this extreme case, the contribution of erosional unloading to the total uplift rate does not exceed 35%.

2. Better justification of the used models.

- a. Both reviewers had questions concerning the ice modeling and why we did not use the LGM reconstruction by Bini et al. (2009). In our literature search prior to the modeling, we did find the reconstruction by Bini et al. (2009), but it covers Switzerland only. No equivalent reconstructions exist for France, Italy, Germany, and Austria. In addition, the documentation that accompanies the publication by Bini et al. (2009) does not describe in detail how the reconstruction was done. Although Bini et al. (2009) stated to have used the same ice-extent indicators (terminal moraines and trimlines) as we did, we do not know how they constrained the ice thickness in the foreland where no trimlines exist. For these reasons, we chose to use a physically based model. Nevertheless, we compared our reconstruction with the existing reconstruction of Bini et al. (2009) and found only an 8% difference in ice volume where the reconstructions overlap, which would only slightly influence our final results.
- b. Reviewer #2 pointed out that there exists a reconstruction of the valley fill thickness in the region around Bern. Compared to our valley-fill estimates, we find that the two reconstructions are fairly consistent (29 km³ and 24 km³, respectively).

- c. Reviewer #2 acknowledges the usefulness of the thin-plate approach for determining flexure of the lithosphere, but asked how deeper-seated, e.g., mantle processes might modulate the isostatic response to loading/unloading. This is a very good point and we added to our discussion the role of mantle processes, particularly with respect to the excess uplift in the Swiss Rhône Valley.

Reviewers' comments:

Reviewer #1 (Remarks to the Author):

Review of "Glacial isostatic uplift in the European Alps" by Mey et al.

In this paper, the authors propose that the current uplift of the Alps is due to viscoelastic response of ice unloading since the last glacial maximum. To do this, they use a reconstruction of the glacial overdeepening, an ice flow model and flexural model to explain the available levelling data. The paper is nicely written, clear and includes thorough calculations. I think it may eventually be published in Nature Communications. However, I have several concerns that I think should be addressed before recommending publication.

First and foremost, I am really concerned about the authority used to express the conclusions that all the uplift must solely be explained by the Earth's viscoelastic response to ice unloading, at the expenses of erosion or tectonics. The authors' analysis is interesting, but it does include a range of poorly constrained parameters. More importantly, the study does not disprove the proposition that about 60% of the uplift can be explained with a flexural model, and all of it with local isostasy only (Champagnac et al., 2009). The Alps are currently eroding (e.g., Hinder, Champagnac, Wittmann, to cite a few), so there must be a response. How much is debatable, but arguing that there is no uplift due to erosion is equivalent to saying that flexural (and to some extent local) isostasy does not work. Maybe I am misunderstanding, for example about how the erosion correction that is done, but the abstract statement is very strong: "we show that virtually all of the geodetically measured surface uplift in the European Alps can be explained by the Earth's viscoelastic response to ice unloading after the last glacial maximum (LGM)". So if the authors want to keep this stand, they should demonstrate convincingly that erosion does not cause any uplift.

We agree that in the original submission, our discussion of how important erosional unloading could be was too brief. In the revised manuscript, we approach this question as follows. First we compare our derived postglacial erosion rates with data from modern river loads (Hinderer et al. 2013), cosmogenic nuclides (Wittmann et al. 2007; Norton et al. 2010,2011), and thermochronology (Fox et al. 2015), which integrate over timescales of 100 yrs, 1000 yrs and 2 Myrs, respectively. We show that for six closed basins (i.e., where rivers traverse lakes as they leave the Alps), our mass budget based on valley fills yields comparable erosion rates, which are rather on the higher end of reported rates. Our estimate of the postglacially eroded mass is thus near the upper limit of probable values. Second, we assumed that 10% of the post-glacial eroded material was exported and 90% was trapped (see Fig.8 in Hinderer, 2001). Next we computed the lithospheric response to the erosion/deposition and directly converted this into an uplift/subsidence rate, similar to what Champagnac et al. (2009) did, and thereby accounted for the fact that the Alps are continuously eroding. We calculated the postglacial rebound for different EETs and for different mantle viscosities

and then added the respective erosional component. Thus, we treated the processes of deglaciation and erosional unloading differently, with the former being time-dependent and the latter being constant since 17 ka BP. We found that the erosional contribution to total uplift was ~10%. Following those calculations, we repeated our analysis for an extreme (unreasonable) case in which 90% of the eroded material was exported and only 10% trapped, for illustrative purposes. In this extreme scenario, the erosional component of the total uplift rate increases to a maximum of 35%. Hence, we now feel quite confident that the erosional component to total uplift is rather small, and we modified the above sentence in the abstract to better reflect the results of our modeling: *"Here we show that ~90% of the geodetically measured rock uplift in the Alps can be explained by the Earth's viscoelastic response to ice unloading after the LGM."* (new manuscript P1L16)

Second, the use of ice flow model should be better justified. There exists an ice reconstruction (Bini et al., 2009) that could be used, so I did not fully understand why the authors needed to run models, especially that the climate part is poorly constrained.

In our literature search prior to the modeling, we did find the reconstruction by Bini et al. (2009), but it covers Switzerland only. No equivalent reconstructions exist for France, Italy, Germany, and Austria. In addition, the documentation that accompanies the publication by Bini et al. (2009) does not describe in detail how the reconstruction was done. Although Bini et al. (2009) stated to have used the same ice-extent indicators (terminal moraines and trimlines) as we did, we do not know how they constrained the ice thickness in the foreland, where no trimlines exist. For these reasons, we chose to use a physically based model. Nevertheless, we compared our reconstruction with the existing reconstruction of Bini et al. (2009) and found only an 8% difference in ice volume where the reconstructions overlap, which would only slightly influence our final results.

The reviewer is right that the climate during the LGM is poorly constrained. However, as it is not our goal to reconstruct the paleoclimate from the ice extent, we don't think this is a problem. Our main goal is to have a good estimate of the ice mass, which is the relevant point for determining the lithospheric deflection. As we explain in the methods section, we iteratively adjusted the equilibrium line altitude (ELA) by catchment until the modeled ice extent approximately matches the mapped ice extent. It is interesting to note that the resulting map of ELA lowering during the LGM shows both north-south and west-east gradients, reflecting what we would expect from a glacial climate. Given a particular ice extent, the equilibrium geometry of the ice cap is mostly governed by the ice dynamics and less so by the climate. In this regard, we think that our physically based approach is preferable to the geometrical reconstruction by Bini et al. (2009), because it is mechanically consistent with our understanding of ice physics.

Furthermore, the modelled ice volume is different than the reconstructed one. That must affect the calculations.

We cropped our Icecap reconstruction to the extent presented in Bini et al. to allow for better comparability. It appears that our modelled ice volume is 27,000 km³ and the value derived from the Bini map is 25,000 km³, which is a difference of only 8%. Provided all other uncertainties and insufficient constraints to judge which of the reconstructions is more accurate, we deem these differences negligible.

A comparison of modelled rock uplift rates between the icecap reconstruction and the modelled icecap would be useful to assess whether it matters. It would definitely justify the model better.

We agree that comparison of how different reconstructions affect model predictions would be useful to assess the sensitivity of our approach. However, as stated above, the Bini et al. (2009) map covers only a part of the LGM icecap and we were interested in the lithospheric response to the entire icecap. Unless the effective elastic thickness is low and quasi local isostasy applies (something that we argue against), calculations with only the Swiss part of the iceload would lead to an underestimation of the deflection. In any case, our modelled ice volume is only 8% higher than the one derived from the Bini map, which we think is in good agreement. Nevertheless, we repeated our calculation and replaced our iceload in the Central Alps with the one from the Bini et al. (2009) reconstruction, and found that our conclusions do not change (see also response to the respective comment by reviewer #2).

Third, the final discussion is very disappointing. The authors talk about specific aspects about the Alps. Instead, or in addition, they should discuss the implications of their claims to our understanding of mountain ranges in general.

We thank the reviewer for being clear about this important aspect and we added another paragraph on the general implications. *“Although the Alpine LGM-icecap was much smaller than the northern hemisphere icesheets, the Earth’s response to its disappearance dominates the vertical motion over broad regions of the Alps even today. Other mountain belts that were similarly affected by glaciation, such as the Alaska Range, the Himalaya, or the Southern Alps in New Zealand, might therefore also exhibit such a long-wavelength uplift component.”* (new manuscript P9L215).

Line by line comments:

Line 47: change 'furnish' by 'provide'

Done

Line 57-60: It would only be fair to mention some of the recent papers that gave constraints on the timing of increase erosion in the Alps, as it is given for the other aspects of the Alps geological history. There were several papers in recent years (e.g. Veron, Glotzbach, Valla, Fox to cite a few). We agree and added the following sentence, in which we cite some of the above mentioned papers: *“Locally, glaciation was presumably associated with a two-fold increase in exhumation rates^{18,19} and topographic relief²⁰, which may be controlled by feedbacks between glacial erosion, crustal unloading, and isostatic uplift.”* (new manuscript P3L59)

Line 75-76: using 'surface lowering' is confusing here. It seems it does not include isostatic response to erosion. Please clarify this statement.

We agree and replaced the old sentence *“By distributing this volume evenly over the Alps (~123,000 km², excluding valley fill area) and assuming a mean density of 2,000 kg m⁻³ for unconsolidated sediments and 2,700 kg m⁻³ for bedrock, we compute a total surface lowering of 11.9 ± 1.3 m.”* with the following one: *“By distributing this volume evenly over the Alps (~123,000 km², excluding valley-fill area) and assuming mean densities of 2,000 kg m⁻³ for unconsolidated sediments and 2,700 kg m⁻³ for bedrock, we computed a total eroded mass of ~4×10³ Gt, which would correspond to a rock column of 11.9 ± 1.3 m.”* (new manuscript P4L89) to clarify the statement.

Line 87-88: Please explain better why there is such a large difference in ice thickness. What is correct, the reconstruction or the ice thickness? What is the effect of such difference on the rebound?

Because in previous reconstructions, the thickness of the valley-fill deposits were poorly constrained, Florineth (1998), Florineth and Schlüchter (1998) and Kelly et al. (2004) focused on reconstructing the ice surface elevation rather than the former ice-thickness. Kelly et al. (2004) noted that: "...the bedrock floor of the Rhône Valley at Martigny is approximately 1000 m below the present valley floor (Pfiffner et al. 1997)... Therefore, the LGM ice thickness near Martigny may have been as much as 2300 m." Their assessment is not far from our modelled ice-thickness at this location of 1800 m. Because the additional ice volume that is due to our removal of the valley-fill deposits (1800 km³) is small compared to the total ice volume (68,000 km³) we suspect only a small effect on the rebound, particularly if the loads are more regionally compensated.

Line 125: Same comments as for line 57-60. There are thermochronometric studies do not suffer for the incomplete preservation (though they suffer from other problems...).

We added two sentences to take into account the thermochronometric studies: "*Several thermochronometric studies point to an increase in exhumation rates of similar magnitude over the last 2 Ma, which has been attributed to a positive feedback between glacial incision, isostatic rebound, rock uplift and exhumation rates^{19,20}. However, we argue that much of the material eroded from the Alps since deglaciation was deposited within Alpine valleys and therefore does not contribute to erosional unloading.*" (new manuscript P7L166)

Line 126-127: Point well taken, but that does not disprove the alternative option that erosion causes the uplift. It is not because your model fits the data that it disproves the other option.

Please refer to our main response in the beginning of this document. We directly calculated the erosional component to total uplift for both a best-estimate and extreme scenario, and we find that in both cases, the erosional component is relatively small (10% in the best-estimate scenario, and 35% in the extreme scenario). We have also added arguments to the text noting that if other processes contribute to uplift, they must do so in a pattern that is very similar to the pattern predicted by glacial-isostatic adjustment, which we believe is unlikely.

Line 130-144: See main comment.

See answer to main comment at the start of our response letter.

Line 188: remove "the" before Poisson

Done

Line 241: change "inverted" by "optimized"

We prefer to keep the term "*inverted*" because "*optimized*" would change the meaning of the sentence.

Reviewer #2 (Remarks to the Author):

The authors of this paper claim that the current uplift of the European Alps, measured with geodetic techniques, is the response to glacial unloading since the termination of the LGM c. 17 Ka ago. They used a 2D elastic thin-plate model overlying a viscous mantle as a proxy for modelling the response of the European lithosphere to unloading, and an inferred thickness pattern of LGM ice derived through the use of a numerical ice flow model to solve the shallow ice approximation. The authors also performed a sediment budget to account for the contribution of Holocene erosional unloading to the currently measured uplift. The authors indeed address a problem that has not been solved during the past decades, and as such any improvements in this field will be of great significance. I am much in favor of seeing this topic addressed, discussed and results being published, particularly in this journal that is designed for a broad readership, but I do see space for improving the current manuscript. The major points include: (i) a better explanation of the novelty of this research, (ii) a better justification of the selected lithospheric model and ice flow model, (iii) a more careful treatment of how published data is being handled, and (iv) a more careful consideration of the data and processes in the Eastern Alps.

We thank the reviewer for the positive appraisal of our study and for making valuable suggestions for improving the manuscript. We hope the reviewer finds that our revised version addresses all of the above points and stands stronger than before.

Let me first outline my first point, and this addresses the novelty of the presented manuscript. Gudmundsson (1994, paper cited in this paper) addressed the same question for the Central Alps, used a similar approach and came to the same conclusions, both regarding the glacial unloading as a major driving force, the inferred viscosity of the underlying mantle, and the deviations between model and observation particularly in the Valais.

The reviewer is right that Gudmundsson (1994) already addressed this question, which is, as the reviewer rightly highlights, a long-standing puzzle. Gudmundsson (1994) used an analytical approach with the icecap being represented by a cylinder with a radius of 150 km and a thickness of 250 m. The resulting volume of this cylinder is $\sim 18\,000\text{ km}^3$, which is much smaller than our reconstructed ice mass and also less than half of the value reported for the LGM ice volume in Table 1 of the same paper ($\sim 40,000\text{ km}^3$). Gudmundsson notes at the end of his paper: *“In view of the uncertainties involved with the estimates done so far, a more promising approach seems, however, to be a true 2D-modelling where the spatial distribution of the Late Würm glaciation would be accounted for. The Alpine glaciation during Würm did not have the form of an ice sheet, and the form of the load used can only give an order of magnitude estimate.”*

He further states: *“The conclusion that a glacial isostasy can, ... give rise to uplift-rates comparable in magnitude to the measured ones, stresses the need and can be seen as a justification for the development of a model with a more sophisticated load distribution.”*

Therefore, expanding on the first-order estimate and resolving the spatial pattern of glacial rebound enables us to not only compare uplift-rate magnitudes but also their spatial distribution. Because we are also considering erosion-driven uplift, we think that this can indeed be regarded as a step towards deconvolving the individual uplift components. In our revised manuscript, we expanded on

this point more specifically and compared spatial patterns of uplift as induced by spatially variable ice and erosional unloading to geodetically measured ones. This way, we make use of the improved spatial resolution of our model and obtain an important handle for comparing model results with observations that was previously not possible.

We would also like to note that although the values for the upper mantle viscosity might be similar (they are actually lower in our study), the EETs are not. While Gudmundsson (1994) achieved the highest rates (1.4 mm/yr) with an EET of 11 km (Fig.4), we argue that the EET is most likely much higher at ~50 km. This discrepancy is probably due to the different assumptions for the ice mass (see above) and the different timing of deglaciation (13,000 yr BP vs. 17,000 yr BP).

In summary, we hope that we managed in the revised version to both make better use of our novel modeling approach and manage to better convey the novelty to the reader.

In the same sense, also applying a similar modeling approach, Champagnac et al. (2009, cited in this manuscript) came to different conclusions in the sense that most of the uplift we currently see in the Central Alps is due to erosional unloading. Champagnac et al. also detected deviations between geodetic uplift and their model particularly in south, similar to the here presented study. The authors of this manuscript mention this and say that the LGM ice load was much larger than the post-glacially eroded sediments. They then refer to their own model that suggests the dominance of ice melting over erosional unloading. I have not been satisfied by this explanation as it does not clarify the situation. This then means that the differences between the conclusions by Champagnac et al. and this paper remain non-discussed.

This important point was also raised by reviewer #1 and we have addressed it in much more detail in the revised version. Please see our general answer #1 at the beginning of this document and our answer to the first comment of reviewer #1.

In the same sense, it is not clear to me how much of the geodetic uplift is still related to the geodynamic processes operating at much larger time scales (e.g., Baran et al., 2014, *Lithosphere*; Fox et al., 2015, *Geology*; Wagner et al., 2010, *EPSL*). For instance, how much of the uplift measured in the Eastern Alps is due to the rotational collision of the Adriatic plate? I do acknowledge that the authors focus their analyses on the Holocene and the modern situation, but some of the modern uplift (if not all?) could also be related to long-term processes including slab breakoff (beneath the Western Alps) and intra-crustal delimitation (papers cited above), and clockwise rotation of the Adriatic plate with the potential to shorten the Eastern Alps. If the authors have sorted out these questions and still conclude that most of the modern uplift is due to glacial unloading, then this paper would represent a breakthrough in our understanding of the Alps. In summary, I see the need that the authors: (i) explain the differences between their work and the results by Gudmundsson (1994), (ii) properly discuss why their conclusions should be more valid than those by e.g., Champagnac et al. (2009), and (iii) disentangle between the long-term and short-term controls/mechanisms of uplift.

Please see our answers to points (i) and (ii) above. Regarding point (iii), we agree that a more thorough discussion of these aspects (as also asked for reviewer #1 is useful and would improve the manuscript. We added a paragraph that accounts for the possibility that, e.g., slab break-off (Nocquet et al 2016, *Nature Sci. Report*) or crustal viscosity anomalies (Chéry et al. 2016, *GRL*)

influence the uplift rates in the Western Alps and concede that this might also be the reason for excess uplift in the Rhône Valley.

My second concern addresses the selection of the models. I do appreciate that thin-plate approaches have been conventionally been used for modeling the deflection of the European continental plate, and I admit that I have done the same before. I am still convinced that this is a valid approach for the Scandinavian lithosphere that extends over thousands of kilometers. However, collisional orogens such as the Alps have a much different lithospheric architecture that comprises (for the central Alps): (i) the c. 1.5 km-thick topographic load made up of upper crustal rocks, (ii) the c. 50 km thick crustal root made of upper and lower crustal material; the crustal root is buoyant and explains most of the negative Bouguer gravity anomalies of c.180 mGal in the Central Alps (Lyon-Caen Molnar, 1989; Holliger & Kissling, 1992), plus (iii) the subducted c. 160 km-long lithospheric mantle that yields a downward directed slab load force (Lippitsch et al., 2003, also cited in the manuscript). Accordingly, while a c. 50 km thick lithosphere that extends over thousands of kilometers can be modeled with confidence by the thin plate approach, the dimensions in the Alps (>50 km thick crustal buoyant root beneath the core of the Alps plus the 160 km-long subducted lithospheric mantle) makes this justification more difficult. Besides, the slab hangs nearly vertically beneath the Bavarian Alps (Lippitsch et al., 20013), suggesting nearly zero mechanical strength of the plate, at least in this part of the Alps. Contrariwise, the slab dips with only 60 degrees beneath the Central Alps. This documents the occurrence of some remaining flexural rigidity in the European lithosphere. Beneath the Western Alps, the slab has possibly broken off (Lippitsch et al., 2003). This yields a complex 3D architecture that most likely imprints on the most recent uplift and exhumation history (Baran et al., 2014; Fox et al., 2015). The authors account for spatial variations of the elastic properties of their model plate, without however explicitly referring to the results of geophysical and seismo-tomography surveys. My point is that the deep structure beneath the orogen should not be discarded (Moho, crustal root, subducted or broken slabs etc.) and needs to be more carefully discussed. We have presented a simplified sketch of this in Schlunegger and Kissling (2015), based on which either the simplest case of Airy-Heiskanen isostasy could be applied where the local accommodation of crustal thicknesses conditions the topographic heights. Alternatively the more complex case of Vening Meinesz isostasy where loads are regionally compensated might be more appropriate. I do acknowledge that I have a biased view on this since I have worked with these models recently. But whatever approach is the most appropriate for collisional orogens: The authors might be correct in selecting their preferred model, but this warrants a better justification particularly considering the architectural complexity of collisional orogens.

We appreciate this detailed comment and agree that our representation of the alpine lithosphere is a simplification. For the calculation of the load-induced deflection, we need to have an estimate of the flexural rigidity of the lithosphere, which is represented by the effective elastic thickness (EET). We see the reviewers point that deep structures might influence the deflection pattern and hence the uplift. However, we started our work with the most simple model of the lithosphere, which is an elastic plate underlain by a viscous substrate. Because this simple model does a good job in explaining the observations, we decided against adding more complexity, which in any case is a difficult task. There currently exists no widely accepted lithospheric model of the European Alps that translates results from tomography studies, e.g., density anomalies, to lithospheric strength properties. Nevertheless, we agree that deep structures may matter, and we have taken care to better address these points in the revised manuscript.

Specifically, we think that slab break-off and asthenospheric upwelling in the Western Alps may lead to uplift rates that exceed the combined signal from erosion and deglaciation, and added the following text to the revised version: *“Because the EET only reflects the flexural properties of the lithosphere, it does not account for deeper-seated processes, which potentially modulate the isostatic response to loading and unloading. We address these aspects in the discussion.”* (new manuscript P6L135) and further: *“Recent analysis of geodetic data in the Western Alps has revealed a narrow zone of high uplift rates of 1.5–2.5 mm yr⁻¹, which exceed the combined signal of GIA and erosional unloading by up to 1.5 mm yr⁻¹[22] (Supplementary Fig. 7b). This excess uplift may be attributed to a contrast in crustal viscosities between the foreland and the mountains, which could enhance the isostatic response to crustal unloading⁴⁵. Because the zone of high uplift coincides with a low P-wave-velocity anomaly at 100–150 km depth¹⁰ and high Quaternary exhumation rates¹⁸, it may also (or alternatively) be related to deep-seated mantle processes that act on million-year time scales²². We thus speculate that the excess rock uplift we observe in the Swiss Rhône Valley may be linked to similar processes.”* (new manuscript P8L192)

In the same sense, the selection of the shallow ice approximation for reconstructing the ice cap needs to be better justified. Related equations have been derived for thin ice, where the ice surface and the bedrock topography at the base of a glacier are nearly parallel to each other (approach selected in this paper). In this case, the basal shear stress τ_b has been related to the ice thickness and the gradient of the ice surface. However, this model does not consider processes such as dislocation creep and recrystallization processes within the ice, driven by pressure gradients of the overlying ice. These mechanisms turn thick ice into a viscous material and could play an important role particularly when explaining the velocity fields of thick trunk glaciers, and I guess that this was the case during the LGM. The authors justify their simplifications because it allows them to compute more efficiently the buildup of the ice cap, which is a fair compromise. However, one problem with the thin ice approach still remains, and this lies in the misfit between the ice surface and bedrock topography particularly when comparing the Bini et al. (2009) map and the current landscape of the Central Alps. The two surfaces are not sub-parallel to each other, particularly for regions surrounding current lakes. Finally, the use of recent annual precipitation does not account for the inferred S-derived moisture sources during the LGM, as proposed by Florineth and Schlüchter (2000). In summary, I propose the author to better explain the preferred selection of their ice growth model along why the current precipitation pattern serves as a proxy for the LGM times, although other authors proposed that atmospheric circulations were different in the past (e.g., Florineth and Schlüchter, 2000).

The reviewer raises the question of whether the ice model based on the shallow-ice approximation (SIA) is appropriate for this study. We acknowledge that the SIA is a simple approach that may not be appropriate when determining the pattern and velocity of ice flow in steep terrain or for studying glacial erosion (see discussion in Egholm et al., 2011). In our study, however, we were only interested in the steady-state ice geometry that matches the ice extent and thickness indicators. Leysinger Vieli and Gudmundsson (2004, JGR ES) investigated the effect of the SIA-simplifications on the steady-state glacier length and found that differences compared to a full-Stokes model that takes all additional stresses into account, are minor. For estimating the ice load on the lithosphere, we thus think the chosen approach is reasonable. To better justify our approach within the manuscript, we added the following sentence: *“We acknowledge the limited explanatory power of the SIA-approach*

concerning the calculation of the ice velocity and flow patterns in steep terrain. The GIA however, is ultimately controlled by the ice geometry and is insensitive to ice velocity and the flow pattern. Compared to higher-order models, only slight differences in glacier geometry are expected during steady-state⁴⁸.” (new manuscript P10L241)

We are aware that the present-day precipitation pattern that we used is most likely different from that of the LGM. Although there exist arguments for an increased southerly moisture source during the LGM, translating such aspects into a realistic LGM map of precipitation without using a paleoclimatic atmospheric model, which is beyond the scope of this study, is difficult to do and likely to receive similar criticism. Instead, we argue that the source of moisture and the exact pattern of LGM precipitation are not relevant for our study. We agree that there may be more realistic reconstructions of the LGM climate and that there may be many combinations of precipitation, ELA and mass balance gradients that would lead to the same ice geometry that we reconstructed. Our main argument is that as long as we succeed in fitting the trimlines and the ice margins, we probably have a good representation of the LGM icecap in terms of its volume and mass, which is the foremost important aspect dictating the lithospheric flexure.

As the third point: I am wondering why the authors have not taken the ice cover map by Bini et al. (2009) to constrain their LGM ice model, at least for the Central Alps. The map has been reconstructed based on detailed observations of trim-lines in the Central Alps and considering available local maps. It is available from Swisstopo upon request and can be easily transferred into an ice thickness map. In this context, I am wondering why the rebound effect should not be much larger in the foreland given the relatively thick LGM ice there (e.g., nearly 1000 m-thick LGM ice above lake Geneva, and c. 700-800 m above Lake Constance). In either case, I strongly recommend using the Bini et al. (2009) LGM reconstruction to constrain ice growth models, else justifications/explanations are needed.

That is a good point that was also raised by reviewer 1. Please see also our response above. Following the suggestion of the reviewer and replacing our ice thickness model in the Central Alps with the reconstruction of Bini et al. (2009), the resulting uplift rates, errors, and best-fit upper mantle viscosity are only slightly different from our original reconstruction. We thus justified our model by adding the following sentences to the manuscript: “*When compared to the existing reconstruction of the LGM ice cover in the Central Alps³² ($V = 25,000 \text{ km}^3$), our modelled ice volume is only ~8% larger ($27,000 \text{ km}^3$), which has negligible effects on our final results.*” (new manuscript P5L112)

In the same sense, I have been wondering why the authors have not used existing bedrock topography models to constrain their estimation of valley fill thicknesses. In particular, there exists a bedrock topography map compiled for the Swiss Alps by Dürst Stucki and Schlunegger (EPSL); the data is also available upon request from Swisstopo. This map is based on a compilation of thousands of drillings and has a yet unbeatable resolution, at least to my perception.

We followed the reviewer’s suggestion and obtained the bedrock reconstruction of the Bern area from Swisstopo to compare it with our estimates. When cropped to the extent within the mountains, i.e., where both models overlap, the sediment volume of this reconstruction is 24 km^3 whereas our

model has a volume of 29 km³. We added a sentence to the relevant paragraph to show that we have considered the existing work: *“Compared to an independent sediment-thickness reconstruction at much higher spatial resolution (10 m) in the region around Bern²³ our reconstruction yields a fill volume of 29 km³, which is ~20% higher than the previous estimate of 24 km³, and thus reasonably consistent when considering the differences in spatial resolution.”* (new manuscript P4L79)

Last but not least, the statement that the valley fill must have formed following ice retreat at c. 17 Ka (line 76) is not correct as the valley fill sediments are older than the Holocene. For instance, some of the overdeepening fills have been dated around MIS 8, or 270 Ka (Preusser et al., 2010; 2011; Dehnert et al., 2012). Older ages for the Thalgut site of 420 Ka have been reported by Preusser et al. (2011). Therefore the erosional budget presented here needs to be corrected. Since the authors claim that the contribution of erosional unloading is minor, the correction will not alter the main conclusion.

We thank the reviewer for pointing to these references and change our statement to: *“With the assumption that the valley fills have formed following ice retreat ~17 kyr BP²⁴, this eroded column corresponds to a mean postglacial denudation rate of 0.7 ± 0.08 mm yr⁻¹.”* (new manuscript P4L92) and later: *“Furthermore, because there might be also older sediments that predate the LGM³⁰ incorporated in our valley-fill estimate, we think that our estimate of postglacial erosion and sediment redistribution is at the upper limit of probable values.”* (new manuscript P4L99)

The important point here is that our estimated postglacial erosion rate is at the higher end of probable values, which is supported by comparison with previously reported erosion rate estimates that we have compiled and added to the revised version. As a result, our estimate of the potential contribution of erosion to present-day uplift should be considered a maximum.

Finally, one needs also to consider that valley fills do not account for the solute loads, which might be up to 50% and even more in some cases (see articles by Hinderer and colleagues). This is the major reason why the 10Be-based basin averaged erosion rates might be more suitable for estimating the amount of erosion unloading at least during the Holocene (Champagnac et al., 2009) as this methodology includes unloading through physical and chemical erosion (Wittmann et al., 2007).

This is a very good point and we explored this effect in more detail in the revised version. We agree with the reviewer that our approach does not explicitly account for the dissolved load. Hinderer (2001) compared post-LGM valley-fill volumes with extrapolated chemical fluxes, suggesting that the exported volume from the Rhône and Rhine catchments is due to dissolved load and correspond to only 5% and 8% of the valley-fill volumes, respectively. In the temporarily closed Inn catchment the exported volume is equivalent to 16% of the valley-fill volume, with half of that being dissolved load. In our estimate, we account for the dissolved load by adding 10% exported material volume to all our valley-fill derived postglacial erosion rates. Moreover, we compiled erosion rate estimates based on gauging data, cosmogenic nuclides, and thermochronology. In comparison with erosion rates based on our valley fills, all other methods yielded similar but generally lower erosion rate estimates. We therefore conclude that our estimates of postglacial erosion should be considered a maximum.

As a forth point, not much is said about the Eastern Alps and the Western Alps (this actually also concerns my review, which has a strong bias towards the Central Alps). The authors claim to present

a view across the entire Alps, but the discussion mainly focuses on the central portion. For instance, Wagner et al. (2011) thought that effects from the counterclockwise rotation of Adria could be seen by the young uplift of the Styrian Basin in the eastern Alps. This rotation might have resulted in crustal shortening and underthrusting of the Pannonian fragment. There is quite a long list of papers that have discussed the most recent tectonic processes in the Eastern Alps the authors need consider (Wagner et al., 2010; 2011; Wölfler et al., 2012, and others).

We admit that our original manuscript was more focused on the Central Alps. After having read the papers cited by the reviewer and several others, it appears that the tectonic component is small compared to the geodetic data. However, we acknowledge that in the regions of high seismicity (Friuli and Lombardy), the tectonic uplift component may be enhanced. Unfortunately, we do not have uplift measurements for these regions and can only speculate about the drivers of uplift. We added to the discussion: *“In the Eastern Alps, because of the ongoing N–S convergence, we might expect a larger tectonic contribution to the recent uplift rates in this region. Estimated rock uplift rates in the Tauern Window (Fig. 2) and along the eastern edge of the Alps since 4 Ma are $<1 \text{ mm yr}^{-1}$ [9] and $0.1\text{--}0.15 \text{ mm yr}^{-1}$ [46], respectively, and thus one order of magnitude lower than geodetically observed uplift rates. Adding this tectonic component to our modelled uplift rate would in both cases improve the fit with the measurements.”* (new manuscript P9L202)

In summary, this has the potential to be great and heavily cited paper addressing a topic that currently receives much attention, but existing literature need to be better incorporated, selected models better and more carefully justified, the eastern and western Alps need more attention, and long-term geodynamic controls on uplift needs to be disentangled from short-term effects.

Again, we thank the reviewer for all the constructive comments and hope that we sufficiently addressed the above concerns in the revised manuscript.

Kind regards

Fritz Schlunegger
Bern, 27th of March 2016

Reviewer #3 (Remarks to the Author):

Overall, I thought this was a very strong paper and have no major qualms with its publication as is. It is clear that a lot of work has gone not only into the research in the paper but also into the writing of the paper, as no typos or other small errors were noticeable. Overall, this is a very well written paper that successfully documents and describes the interesting study that the authors have undertaken.

Here are my organized comments based on the organized points:

A. Summary of the key results

Previous work on the Alps has generally been inconclusive on the specific cause of the modern patterns of geodetically measured surface uplift or has often attributed the uplift to the redistribution of sediment due to glacial erosion and deposition. Using a combination of paleoglacial

modeling, viscoelastic modeling of the lithosphere and upper mantle, and redistributing the valley fills back up to higher elevation, this current study seeks to determine the dominant process in creating the modern and observed surface uplift. By varying properties within realistic or specific ranges, such as the elastic properties of the lithosphere and the glacial sliding coefficient, the authors were able to reasonably match modern measured uplift, valley fill thickness, and glacial trimline data. From this, they concluded that, for the most part, measured uplift in the Alps can mainly be attributed to isostatic rebound due to the lost off icecaps since the LGM, with local strong tectonic components in some isolated regions.

B. Originality and interest

This research, while building off of previous work surrounding uplift in the Alps, certainly stands on its own as a new approach to better understanding the interplay of different processes in this and similar mountain ranges. This work should be of interest not only to those who particularly interested in the Alps but also researchers who generally study sediment sources related to glacial erosion and landscape formation.

C. Data & methodology

In general, this is a novel approach to understanding uplift in the Alps. Only recently, has there been the ability to effectively combine several types of models (for example, lithospheric flexure and glacial flow) along with topographic/sedimentary redistribution. This research certainly takes advantage of the state-of-the-art techniques in these fields, while also exploiting the existing measurements of sediment depth and high precision geodetics.

As for the glacier model, there are some minor criticisms that likely only come to mind due to the length constraints of a Nature Communication paper. While the EET and lithospheric/mantle rheology variables are chosen based not only on reducing the uplift residuals, and upper mantle viscosity is compared realistically to other estimates from different regions, the variables used in the glacial model seem less explicitly informed by reality. This is not too say that they are poorly chosen but just that little justification or comparison is given for variables such as the choice of overburden pressure or the sliding coefficient. In fact the definition of the sliding coefficient, "depending inversely on the bed roughness," makes it seem like this term could in fact be related to the existing topography. While they attribute the foreland and mountain interior differences in this term to the existence or lack of deformable sediments and more meltwater, comparing the actual values and the choice of increase amount to other model results could at least ensure that the glacial modeling is consistent with previous uses of this and similar models.

First of all, we would like to thank the reviewer for the positive assessment of our study. We are glad to provide more details about the ice modeling part. To justify our choice of the effective pressure we added the following to the respective paragraph describing our ice modeling: *"In reality N would be highly variable in both, space and time⁴⁹ with lower values leading to increased decoupling between ice and bed, which permits faster sliding. Tests with N being 80% of the ice overburden pressure indicate only a small increase in the resulting ice volume."* (new manuscript P11L250)

Regarding the contrast in sliding coefficients between the foreland and the orogenic interior, we added a citation to Gregoire (2010), who showed that the Laurentide ice-sheet margins could only be successfully fitted with a 20-fold increase of the sliding parameter in regions where the ice slid over

sediments. We added to our manuscript *“This pattern was reproduced when A_s in the foreland was increased by a factor of 15, which is similar to the value used in a reconstruction of the Laurentide icesheet⁵³.”* (new manuscript P11L265)

D. Appropriate use of statistics and treatment of uncertainties

The only significant statistics in the paper relate to the model fitting, particularly the use of RMS. This is an entirely reasonable and appropriate method for the analysis as shown.

E. Conclusions

By showing the sensitivity of the model in Extended Data Figure 5, the author's emphasize the robustness of their results for their given models. The conclusions stand strongly against the observations. Even the regions of higher RMS are systematic and make sense given the tectonic situation of the Alps.

F. Suggested improvements

The additional information, expanded upon in Section C, that might be beneficial would be a comparison or table (similar to Extended Data Table 2) that compares the glacier flow parameters to those used in other models.

We hope we could convince the reviewer concerning the choice of our model parameters by the above additions to the methods section (see answer to comments of Section C).

G. References

References seem entirely appropriate and nothing else seems necessary.

H. Clarity and context

The abstract is fairly straightforward and properly summarizes the paper. The only suggestion might be to end the abstract on the overall idea of glacial rebound being the dominant mechanism instead of ending on the exceptions (Swiss Rhone Valley, Eastern Alps). All other sections of the paper are clear and appropriate.

We agree with the reviewer and have modified the end of the abstract to finish with a strong positive finding. “Our study shows that even small LGM icecaps can dominate present day rock uplift in tectonically-active regions.” (new manuscript P1L22)

Reviewer #1 (Remarks to the Author):

I have read the revised manuscript and the response to reviews. Overall the authors have made a sincere effort to address the reviewers' comments. I also remain of the opinion that this is a very interesting and thorough study that should be published in Nature Communications.

The authors have satisfactorily addressed the studies of Champagnac et al. (2009) and Wittman et al. (2007), which argued that a large fraction (up to 70%) of the present-day rock uplift rates may be due to erosional unloading of the lithosphere. While the response and modifications are sound, it would be very useful for readers to explain why there is such a large difference. The discrepancy between the two studies is surprisingly large. Given the influence of these papers, one sentence in the main text would be valuable.

Minor comments:

Line 59: The authors cite Gibbard and Ehlers for glaciations in the Alps. A reference to workers in the Alps would probably be more appropriate than this one.

Line 61: The role of deep-seated mantle processes should be mentioned in this sentence since it is discussed later.

Line 100: Remove 'we think'.

Line 169: 18 should be cited along with 19 and 20.

Line 175: It is not clear how the erosion rates derived on 1-2 Myr using thermochronometry is used to state that the erosion rates peak during deglaciation. Please, clarify.

Line 241: Rephrase 'limited explanatory power'.

Reviewer #2 (Remarks to the Author):

The authors have invested an impressive effort in improving the paper. I have now been convinced that among the various possible driving forces (glacial unloading, erosional unloading, deep crustal processes), the controls of glacial unloading on modern uplift is greater than those related to erosional unloading. I am still not fully convinced that lithospheric delamination, a process that could potentially drive rock uplift provided that the Alps have evolved as a rollback orogeny as we have proposed in a recent paper (Schlunegger and Kissling, 2015; Nat. Comm.), can be ruled out. The statements in lines 185-188 are not convincing, mainly because they are not sustained by arguments. In fact, one cannot rule out other processes by saying 'which we think is unlikely'. A proper solution of this problem requires a model that is more complex than the two dimensional elastic thin-plate equation as applied here because of the Alpine complexities (see below), and I admit that such a model is currently not available that accounts for all of these. In fact, as I have written in my previous review, the Alps have a much much more complicated architecture (160 km-long subducted slab attached to the European lithosphere beneath the Central Alps while being broken off beneath the Western Alps etc.) than the Scandinavian sheet where a two dimensional

elastic thin-plate approach would be fully appropriate. I don't expect the authors to fully solve and discuss this problem, but it would be fair to refer to these complexities for the sake of scientific transparency within a couple of sentences. In addition, a statement is missing in the introduction where the scope of the article is specifically been mentioned. I think the paper is then ready for publication after these two points have been addressed.

Kind regards

Fritz Schlunegger, Bern August 30th

Reviewer #3 (Remarks to the Author):

Overall, I feel that the authors effectively addressed the major points that I raised, which were mostly focused on ice dynamics.

The other reviewers clearly had deeper background on major glaciations in the Alps, and they raised many valid points and criticisms related to other literature in the this field. By incorporating more comparisons with existing studies on this area, the authors addressed these important points. Their rebuttal and response to the reviewers satisfied any questions that I had once I had read the other reviews, and I would say that this manuscript is now suitable for publication.

RESPONSE TO REVIEWERS' COMMENTS:

Reviewer #1 (Remarks to the Author):

I have read the revised manuscript and the response to reviews. Overall the authors have made a sincere effort to address the reviewers' comments. I also remain of the opinion that this is a very interesting and thorough study that should be published in Nature Communications.

The authors have satisfactorily addressed the studies of Champagnac et al. (2009) and Wittman et al. (2007), which argued that a large fraction (up to 70%) of the present-day rock uplift rates may be due to erosional unloading of the lithosphere. While the response and modifications are sound, it would be very useful for readers to explain why there is such a large difference. The discrepancy between the two studies is surprisingly large. Given the influence of these papers, one sentence in the main text would be valuable.

We added a paragraph to the end of the introduction where cause for this discrepancy is mentioned. *"We show that most of the postglacially-eroded material was trapped within the mountain belt and did not contribute to erosional unloading as previously suggested⁸."* [new manuscript L73]

Minor comments:

Line 59: The authors cite Gibbard and Ehlers for glaciations in the Alps. A reference to workers in the Alps would probably be more appropriate than this one.

We replaced Ehlers and Gibbard with Hantke (2011), who focusses on the Alps.

Line 61: The role of deep-seated mantle processes should be mentioned in this sentence since it is discussed later.

done

Line 100: Remove 'we think'.

done

Line 169: 18 should be cited along with 19 and 20.

done

Line 175: It is not clear how the erosion rates derived on 1-2 Myr using thermochronometry is used to state that the erosion rates peak during deglaciation. Please, clarify.

We restructured the sentence so that our observation is separated from the interpretation.

"Furthermore, erosion rate estimates based on our valley-fill volumes exceed long-term rates based on thermochronology (Fig. 3d). This could be explained by a peak in erosion rates at the onset of deglaciation due to intensified paraglacial processes that can be expected for a landscape that is adjusting to new boundary conditions." [new manuscript L183]

Line 241: Rephrase 'limited explanatory power'.

We changed the sentence to: *"We acknowledge that the SIA-approach may not be suitable*

concerning the calculation of the ice velocity and flow patterns in steep terrain.” [new manuscript L254]

Reviewer #2 (Remarks to the Author):

The authors have invested an impressive effort in improving the paper. I have now been convinced that among the various possible driving forces (glacial unloading, erosional unloading, deep crustal processes), the controls of glacial unloading on modern uplift is greater than those related to erosional unloading. I am still not fully convinced that lithospheric delamination, a process that could potentially drive rock uplift provided that the Alps have evolved as a rollback orogeny as we have proposed in a recent paper (Schlunegger and Kissling, 2015; Nat. Comm.), can be ruled out. The statements in lines 185-188 are not convincing, mainly because they are not sustained by arguments. In fact, one cannot rule out other processes by saying 'which we think is unlikely'. A proper solution of this problem requires a model that is more complex than the two dimensional elastic thin-plate equation as applied here because of the Alpine complexities (see below), and I admit that such a model

is currently not available that accounts for all of these. In fact, as I have written in my previous review, the Alps have a much much more complicated architecture (160 km-long subducted slab attached to the European lithosphere beneath the Central Alps while being broken off beneath the Western Alps etc.) than the Scandinavian sheet where a two dimensional elastic thin-plate approach would be fully appropriate. I don't expect the authors to fully solve and discuss this problem, but it would be fair to refer to these complexities for the sake of scientific transparency within a couple of sentences.

We added the following sentence to the respective part of the discussion to account for possible effects of the lithospheric complexity.: *“However, we acknowledge that the Alpine lithosphere has a complex architecture⁴⁷ that our elastic thin plate approach may not be able to fully account for.”* [new manuscript L199]

In addition, a statement is missing in the introduction where the scope of the article is specifically been mentioned. I think the paper is then ready for publication after these two points have been addressed.

We added the following paragraph to the introduction: *“In this study, we re-evaluate the effect of GIA on the present-day rock uplift in the Alps while accounting for postglacial erosion, sediment deposition and variations in lithospheric strength. We show that most of the postglacially-eroded material was trapped within the mountain belt and did not contribute to erosional unloading as previously suggested⁸. Instead our results demonstrate that the long-wavelength uplift signal is best explained by the Earth’s viscoelastic response to ice unloading after the LGM. We conclude that present-day uplift rates in other tectonically active and glaciated mountain belts could also carry a component related to LGM deglaciation.”* [new manuscript L71]

Kind regards

Fritz Schlunegger, Bern August 30th

Reviewer #3 (Remarks to the Author):

Overall, I feel that the authors effectively addressed the major points that I raised, which were mostly focused on ice dynamics.

The other reviewers clearly had deeper background on major glaciations in the Alps, and they raised many valid points and criticisms related to other literature in the this field. By incorporating more comparisons with existing studies on this area, the authors addressed these important points. Their rebuttal and response to the reviewers satisfied any questions that I had once I had read the other reviews, and I would say that this manuscript is now suitable for publication.